# Semantic-guided Diverse Decoding for Large Language Model

**Weijie Shi**[1]*, **Yue Cui**[3], **Yaguang Wu**[2], **Jingzhi Fang**[1], **Shibo Zhang**[2], **Mengze Li**[1],
**Sirui Han**[1]†, **Jia Zhu**[4], **Jiajie Xu**[5], **Xiaofang Zhou**[1]†

[1]The Hong Kong University of Science and Technology, [2]MetaX,
[3]Alibaba Group, [4]Zhejiang Normal University, [5]Soochow University

## Abstract

Diverse decoding of large language models is crucial for applications requiring multiple semantically distinct responses, yet existing methods primarily achieve lexical rather than semantic diversity. This limitation significantly constrains Best-of-N strategies, group-based reinforcement learning, and data synthesis. While temperature sampling and diverse beam search modify token distributions or apply n-gram penalties, they fail to ensure meaningful semantic differentiation. We introduce Semantic-guided Diverse Decoding (SemDiD), operating directly in embedding space that balances quality with diversity through three complementary mechanisms: orthogonal directional guidance, dynamic inter-group repulsion, and position-debiased probability assessment. SemDiD harmonizes these competing objectives using adaptive gain functions and constraint optimization, ensuring both quality thresholds and maximal semantic differentiation. Experiments show SemDiD consistently outperforms existing methods, improving Best-of-N coverage by 1.4-5.2% across diverse tasks and accelerating RLHF training convergence by 15% while increasing accuracy by up to 2.1%.

## 1 Introduction

Diverse decoding of large language models (LLMs) aims to produce multiple semantically diverse responses for the same query. High-quality, creative, and non-repetitive multiple responses play a crucial role in numerous fields. In best-of-N strategies, multiple-answer candidates coupled with a verification mechanism (like self-consistency) enable smaller models to surpass larger ones under equivalent computational budgets [1, 2, 3]. In group-based reinforcement learning from human feedback (RLHF) algorithms like Iterative-RLHF [4, 5], RLOO [6], and GRPO [7], diverse decoding facilitates self-improvement through reward-based evaluation of answer groups. Additionally, in seed-based data synthesis, diverse decoding serves as a fundamental component for generating varied training data, thereby enhancing the robustness and generalization capabilities [8, 9, 10, 11].

Current methods like temperature sampling [12], nucleus sampling [13], diverse beam search [14, 15], and code-book [16] create lexical variations, but they often produce semantically similar responses. To ensure semantic diversity among multiple-answer candidates, an ideal approach is to generate numerous candidates, embedding and clustering them semantically, and then selecting high-probability representatives from different clusters. Unfortunately, this is computationally prohibitive in practice, and calls for resource-efficient methods to achieve semantic diversity during the decoding process.

---

*wshiah@connect.ust.hk
†Corresponding authors

39th Conference on Neural Information Processing Systems (NeurIPS 2025).

We propose Semantic-guided Diverse Decoding (SemDiD), a decoding algorithm that generates multiple ($k$) semantically diverse and high-quality answers through three key mechanisms: directional guidance, inter-group repulsion, and probability preference. Directional guidance steers each decoding trajectory toward distinct semantic regions, inter-group repulsion maintains semantic distances between groups, and probability preference prioritizes tokens with higher likelihood to ensure quality. SemDiD runs $k$ groups of beam searches simultaneously, with the first group employing greedy decoding to establish a quality baseline. Despite the conceptual simplicity, effective implementation presents several challenges:

**Diminishing Returns in Probability-Based Quality Assessment:** While probability offers a computationally efficient quality metric, it has significant limitations. Extremely low probabilities correlate with poor outputs, but quality improvements plateau as probabilities increase beyond moderate levels. Rather than aggressively maximizing probability, SemDiD treats probability as a lower bound using a harmonic gain function that adaptively prioritizes quality or diversity based on their relative performance. This ensures both metrics remain above acceptable thresholds while optimizing for overall effectiveness.

**Position and Length Bias in Probability Estimation:** As sequences progress, tokens appearing later typically receive higher probabilities due to increasing contextual certainty, resulting in distorted quality assessment. To address this, SemDiD employs position-based regularization, applying diminishing weights to tokens in later sequence positions while imposing a threshold to limit the influence of excessively high-confidence tokens.

**Balancing Competing Forces:** To simultaneously maintain quality and maximize diversity, SemDiD integrates harmonic mean theory with $\epsilon$-constraint optimization to guarantee that each group's quality score remains above the greedy baseline threshold while pursuing semantic diversity. Within the diversity dimension, SemDiD implements stage-aware transition between directional guidance and inter-group repulsion. During early exploration stages, partial answer embeddings are closely in semantic space, so directional guidance predominates to establish initial diversity. As decoding progresses, weighting gradually shifts toward inter-group repulsion, recognizing that predetermined directional guidance may become suboptimal and potentially limit model precision.

We evaluate SemDiD on multiple benchmarks to test its effectiveness. For Best-of-N, our approach enhances performance across various tasks, with improvements ranging from 1.4% on ARC-Challenge to 5.2% on MMLU-Pro+ with 25 samples. In the RLHF framework, SemDiD enables more efficient exploration of the solution space through the Iterative-RLHF, GRPO, and RLOO algorithms, reaching target performance levels more quickly during training and achieving 1.8-2.1% higher accuracy compared to the best baseline methods. The code is available at https://github.com/shiweijiezero/SemDiD.

## 2   Related Work

Decoding algorithms determine how tokens are selected during language model generation. Diverse decoding methods extend this concept by producing multiple distinct outputs for a single prompt, and can be categorized into parallel sampling approaches and diverse beam search variants.

Parallel samplings generate tokens independently across different decoding paths. Methods such as temperature sampling, top-k sampling [17], nucleus sampling [13], and typical decoding [18] modify the conditional token distribution to increase randomness, but provide no guarantees against duplicate outputs. Arithmetic sampling [16] uses an implicit arithmetic code-book to ensure character diversity. To prevent sampling tokens with excessively low probabilities that might compromise quality, truncation sampling [19] and epsilon sampling [20] introduce cutoff mechanisms based on probability thresholds and distribution shape.

Diverse beam search variants [15, 14] explicitly model interactions between candidate sequences to prevent redundancy, introducing diversity penalties between beam groups like Hamming, Cumulative, and n-gram penalties. Determinantal beam search [21] formulates decoding as a series of subdeterminant maximization problems based on determinantal point processes (DPPs), inherently encoding intra-set interactions to promote n-gram coverage. Stochastic beam search with Gumbel-Top-k trick [22] and Conditional Poisson [23] enable both exploration and principled sampling without replacement from sequence models. While these interaction-based methods improve lexical diversity, they

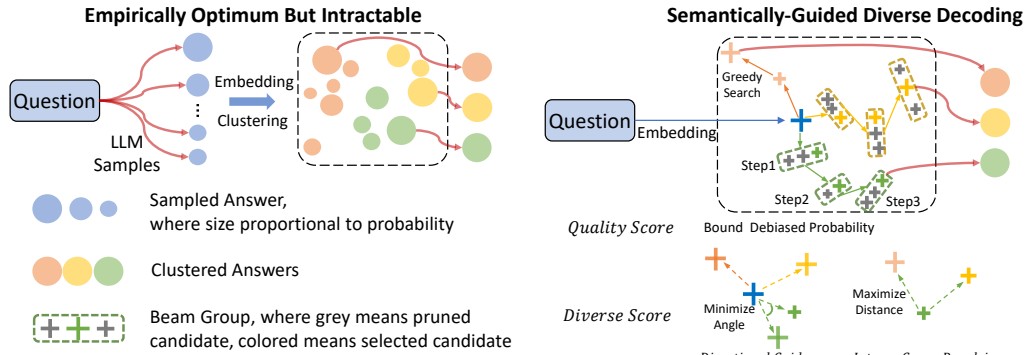

Figure 1: Comparison between ideal semantic decoding and SemDiD. While generating numerous samples followed by embedding and clustering would optimally identify diverse representatives, this is computationally intractable in practice. Our proposed SemDiD achieves similar diversity more efficiently by directly guiding decoding in semantic space through orthogonal directions and inter-group repulsion, while maintaining quality with debiased probability assessment.

primarily operate at the token or n-gram level rather than in semantic space. Our proposed SemDiD advances beyond these approaches by explicitly guiding the decoding process in embedding-based semantic space. Furthermore, to ensure the ease of deployment, our design principle aims to be usable out-of-the-box without any training.

Additionally, recent works have explored semantic diversity in language generation from domain views. [24] use gradient-based attribution scores to identify semantically critical tokens for uncertainty estimation, employing importance sampling with NLI-guided token substitution. [25] propose controlled embedding perturbation at the first token position combined with Bayesian optimization for reasoning in latent space. While these works share our motivation for semantic diversity, SemDiD differs in several key aspects: (1) we operate during beam search decoding rather than post-hoc token substitution or first-token perturbation, (2) our approach balances quality and diversity through harmonic optimization rather than focusing solely on uncertainty or correctness, and (3) we target Best-of-N and RLHF applications rather than uncertainty estimation or reasoning tasks.

## 3 Semantic-guided Diverse Decoding

Let $q$ denote a query input and $M$ a pretrained language model with a conditional distribution $p_M(y|q)$ over possible responses $y \in \mathcal{Y}$. Our Semantic-guided Diverse Decoding (SemDiD) aims to generate $k$ responses $\{y_1, y_2, \ldots, y_k\}$ that exhibit both high quality and semantic diversity.

**SemDiD Architecture.** As shown in Figure 1, SemDiD extends the group beam search framework using $k$ groups and selecting one representative output from each group. Candidates are evaluated on both quality using debiased sequence probability as a lower bound and diversity by assessing embedding space representations through exploration direction and inter-group distance. The first group uses greedy decoding to establish a quality baseline and reference direction.

### 3.1 Efficient Quality Assessment with Debiased Probability

Quality evaluation is crucial during generation. Although dialogue-based LLMs could be used, applying such evaluations to each candidate branch during generation would incur substantial computational costs. An efficient alternative is using the language model's own token likelihood scores, which reflect its confidence level. When a model exhibits uncertainty, it tends to produce lower quality outputs [26].

**Probability as a Quality Lower Bound.** Unfortunately, our experiments [3] reveal that blindly pursuing high probability does not necessarily yield higher quality, as shown in Figure 2. Log

---

[3]Detailed experimental setup is available in Appendix A and B

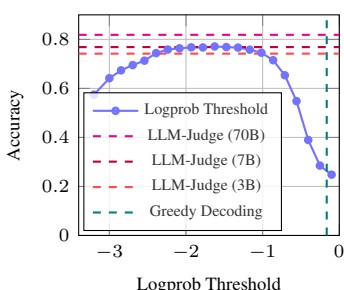

Figure 2: Accuracy in evaluating answer quality.

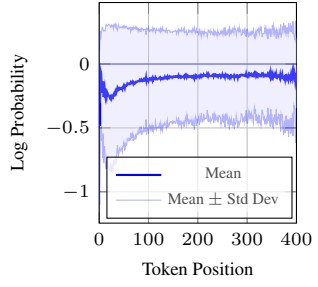

Figure 3: Mean log probability by token position with standard deviation bands.

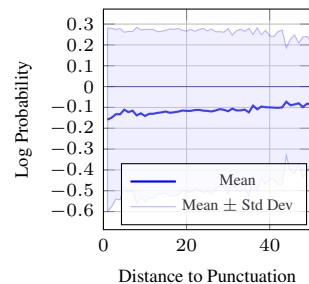

Figure 4: Mean log probability by distance to previous punctuation.

probability thresholds up to -2.5 demonstrate rapid improvements in evaluation accuracy, but this accuracy generally plateaus thereafter. When the log probability threshold exceeds -1, the accuracy of quality assessment significantly decreases, indicating that many correct responses are mistakenly classified as incorrect. This suggests probability-based thresholding is better suited as lower bounds to avoid extremely low-probability sequences and ensure basic quality, rather than as maximization targets for optimal performance.

**Position and Length Biases in Token Probabilities.** Figures 3 and 4 reveal systematic variations in token probabilities that compromise quality assessment. Tokens appearing later in sequences receive artificially inflated probabilities due to richer contextual information. Similarly, probabilities increase as sentences progress after punctuation marks. These position-dependent biases create unfair advantages for longer responses and certain sentence structures. Without correction, probability-based quality assessment becomes unreliable for comparing candidate sequences of different lengths.

We propose a position-debiased probability that applies diminishing weights to token probabilities based on their positions in sequence and sentence:

$$p_{debiased}(t_i|t_{<i}) = p(t_i|t_{<i}) \cdot \exp(-\beta_{seq} \cdot i) \cdot \exp(-\beta_{sent} \cdot d_i) \qquad (1)$$

where $i$ is the absolute position of token $t_i$ in the sequence, $d_i$ is the distance (in tokens) from token $t_i$ to the most recent sentence-ending punctuation, and $\beta_{seq}$ and $\beta_{sent}$ are hyperparameters controlling the respective decay rates. Additionally, we apply a saturation threshold $\tau$ to prevent excessively high-confidence tokens from dominating quality assessment:

$$S_{\text{quality}}(y) = \min(p_{debiased}(t_i|t_{<i}), \tau) \qquad (2)$$

## 3.2 Semantic Diversity Assessment

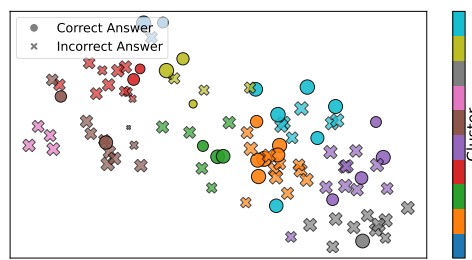

Figure 5: Visualization of 100 answers for the same question, where circle sizes represent probability magnitudes.

Figure 6: Comparison of Best-of-N coverage across highest probability selection, character diversity, and semantic diversity methods.

While probability-based quality assessment provides an efficient mechanism for filtering low-quality candidates, optimizing solely for probability often yields semantically redundant solutions. As

illustrated in Figure 5, correct answers (circles) are scattered across distinct semantic clusters rather than concentrated in a single region. This distribution pattern suggests multiple valid reasoning paths exist in the embedding space.

Furthermore, Figure 6 demonstrates that semantically-driven selection strategies consistently outperform both probability-based ranking and character-level approaches in Best-of-N evaluations. When selecting top-k diverse candidates from 100 samples per problem, semantic clustering achieves up to 95% coverage at k=25, compared to 92% for regular probability-based selection. Surprisingly, we found that embedding model size has minimal impact on performance effectiveness, allowing us to employ smaller, computationally efficient models without sacrificing diversity quality or incurring excessive computational overhead. We employ a small sentence embedding model $\mathcal{E}$ that maps partial or complete sequences to a semantic vector space. Afterward, two key mechanisms drive our semantic diversity:

**Directional Guidance for Initial Exploration:** In the early stages of decoding, embeddings of partial sequences tend to cluster closely in semantic space. This proximity occurs because initial tokens have limited context to differentiate their semantic trajectories substantially. Without explicit guidance, these trajectories would likely converge toward similar semantic regions despite different token selections, resulting in lexical variations with minimal semantic diversity.

To efficiently explore different semantic regions, each group $g$ (except the first greedy group) is assigned a distinct directional vector $\vec{d}_g$ in embedding space. These directional vectors serve as semantic "targets" that guide the decoding process toward different regions of the embedding space. The first group follows the highest probability path using greedy decoding, establishing a reference direction:

$$\vec{d}_1 = \frac{\mathcal{E}(y_t^1) - \mathcal{E}(q)}{||\mathcal{E}(y_t^1) - \mathcal{E}(q)||} \tag{3}$$

where $\mathcal{E}(y_t^1)$ is the embedding of the partial sequence generated by the first group at time step $t$, $\mathcal{E}(q)$ is the embedding of the query. For the remaining groups, we apply the Gram-Schmidt orthogonalization process to generate a set of orthogonal directions:

$$\vec{v}_g = \vec{r}_g - \sum_{i=1}^{g-1} \frac{\vec{r}_g \cdot \vec{d}_i}{||\vec{d}_i||^2} \vec{d}_i \qquad \vec{d}_g = \frac{\vec{v}_g}{||\vec{v}_g||} \tag{4}$$

where $\vec{r}_g$ is a randomly initialized vector, $\vec{v}_g$ is the intermediate vector orthogonalized against all previous directions, and $\vec{d}_g$ is the normalized directional vector for group $g$. The orthogonality ensures each group explores fundamentally different semantic trajectories through the directional score:

$$S_{dir}(y_t^g) = \cos\left(\mathcal{E}(y_t^g) - \mathcal{E}(q), \vec{d}_g\right) \tag{5}$$

**Inter-Group Repulsion for Dynamic Divergence:** While directional guidance establishes initial trajectory divergence, predetermined directions may become suboptimal as generation progresses. Forcing candidates to follow fixed directions regardless of the language model's natural tendencies can lead to suboptimal outputs in terms of quality and coherence. To this end, we introduce an inter-group repulsion that dynamically maintains semantic distance between groups without constraining them to rigid paths:

$$S_{rep}(y_t^g) = -\max_{g' \neq g} \langle \mathcal{E}(y_t^g), \mathcal{E}(y_t^{g'}) \rangle \tag{6}$$

where $\langle \cdot, \cdot \rangle$ denotes the dot product operation. This score becomes more negative as a candidate approaches other groups' semantic regions in the embedding space, thereby encouraging continued divergence throughout the generation process.

The final diversity score for a candidate sequence $y_t^g$ is computed as a weighted combination of directional guidance and inter-group repulsion:

$$S_{div}(y_t^g) = (1 - \alpha_t) \cdot S_{dir}(y_t^g) + \alpha_t \cdot S_{rep}(y_t^g) \tag{7}$$

where $\alpha_t$ is a time-dependent weighting factor calculated as:

$$\alpha_t = \min\left(1, \frac{t}{T_{trans}}\right) \tag{8}$$

Here, $t$ represents the current decoding step, and $T_{trans}$ is a hyperparameter controlling the transition point. During early steps ($t < T_{trans}$), the weight gradually shifts from directional guidance toward repulsion, with repulsion completely dominating after step $T_{trans}$. This dynamic weighting recognizes that directional guidance is more valuable during early exploration stages, while repulsion becomes increasingly important as sequences develop their semantic identity.

### 3.3 Balancing Quality and Diversity

Having assessed quality and diversity, it is critical to optimize these competing objectives.

#### 3.3.1 $\epsilon$-Constraint Quality Guarantee

To ensure the lower bound of quality, we adopt the $\epsilon$-constraint method from multi-objective optimization:

$$\max_{y \in \mathcal{Y}} S_{div}(y) \quad \text{subject to} \quad S_{quality}(y) \geq \epsilon \tag{9}$$

where $\epsilon$ represents our quality threshold based on the greedy decoding baseline:

$$\epsilon = \min_{i \in [1,t]} S_{quality}(y_i^1) \cdot \gamma \tag{10}$$

Here, $y_i^1$ is the partial sequence generated by the first (greedy) group at time step $i$, and $\gamma \in (0, 1]$ serves as a relaxation parameter balancing quality requirements with optimization flexibility.

**Theorem 1 (Quality Guarantee)** *With quality threshold mechanism, the quality difference between any group $g$ and the first greedy group is bounded by:*

$$\mathbb{E}[S_{quality}(y^1) - S_{quality}(y^g)] \leq \delta(1 - \gamma) \tag{11}$$

*where $\delta$ represents the maximum quality dispersion in the response space.*

**Proof:** The first group follows greedy decoding, selecting maximum probability tokens at each step. Through $\epsilon$-constraint, any candidate sequence $y$ with $S_{quality}(y) < \epsilon$ is eliminated regardless of diversity score. Since $\epsilon = \min_{i \in [1,t]} S_{quality}(y_i^1) \cdot \gamma$, and each group must maintain quality scores at least $\gamma$ fraction of the greedy baseline at every step, the cumulative quality difference cannot exceed $(1 - \gamma)$ multiplied by the maximum possible quality dispersion $\delta$.

#### 3.3.2 Harmonic Gain-Based Balancing

While the $\epsilon$-constraint method provides quality guarantees, its binary acceptance/rejection mechanism can be too rigid in practice. In practical applications, we neither want candidates with high probability but low diversity, nor those with low probability but high diversity. To allow more nuanced trade-offs, we employ a harmonic gain function that prioritizes improving the weakest aspect of each candidate:

$$S_{combined}(y_t^g) = \begin{cases} -\infty, & \text{if } S_{quality}(y_t^g) < \epsilon \\ \frac{\lambda \cdot S_{quality}(y_t^g) \cdot S_{div}(y_t^g)}{S_{quality}(y_t^g) + S_{div}(y_t^g)}, & \text{otherwise} \end{cases} \tag{12}$$

where $S_{quality}(y_t^g) = \max(0, S_{quality}(y_t^g) - \epsilon)$ represents the quality surplus above the threshold and $\lambda$ is a hyperparameter that controls the strength of harmonic mean.

**Proposition 2** *The harmonic gain function exhibits diminishing returns for improvements in either quality or diversity when the other metric is significantly lower.*

**Proof:** The partial derivatives of the harmonic gain with respect to quality and diversity are:

$$\frac{\partial S_{combined}}{\partial S_{quality}} = \frac{\lambda \cdot S_{div}^2}{(S_{quality} + S_{div})^2} \qquad \frac{\partial S_{combined}}{\partial S_{div}} = \frac{\lambda \cdot (S_{quality})^2}{(S_{quality} + S_{div})^2} \tag{13}$$

As $S_{quality} \gg S_{div}$, we have $\frac{\partial S_{combined}}{\partial S_{quality}} \approx 0$ and $\frac{\partial S_{combined}}{\partial S_{div}} \approx \frac{\lambda \cdot S_{quality}}{S_{quality}} = \lambda$, indicating that improvements in quality yield minimal gains while diversity improvements are highly rewarded. Conversely, when $S_{div} \gg S_{quality}$, improvements in diversity yield minimal gains while quality improvements are highly rewarded. This adaptive balancing ensures that neither metric is neglected during optimization.

**Theorem 3 (Diversity Guarantee)** *The expected minimum pairwise semantic distance between any two responses generated by SemDiD satisfies:*

$$\mathbb{E}[\min_{i \neq j} d_{sem}(y_i, y_j)] \geq \sigma \cdot \sqrt{\frac{1 - \cos(\pi/k)}{2}} \tag{14}$$

*where $\sigma$ is the semantic space variance and $k$ is the number of groups.*

**Proof:** The directional guidance vectors $\{\vec{d}_1, \vec{d}_2, \ldots, \vec{d}_k\}$ are constructed to be orthogonal in the semantic embedding space. The minimum angle between any two vectors is at least $\pi/k$ radians. Since the cosine distance between two unit vectors with angle $\theta$ is $\frac{1-\cos(\theta)}{2}$, and the expected magnitude of semantic differences in the embedded space is proportional to $\sigma$, the minimum expected pairwise distance is bounded by $\sigma \cdot \sqrt{\frac{1-\cos(\pi/k)}{2}}$.

All scoring components undergo group-based normalization to ensure comparability across different assessments. Detailed algorithm procedures are provided in Appendix C.

# 4 Experiments

We evaluate SemDiD on two categories of tasks: (1) Best-of-N evaluation across a variety of benchmarks to assess diversity and quality of generated responses, and (2) the impact on reinforcement learning from human feedback (RLHF) algorithms to measure training efficiency and performance improvements. We evaluate SemDiD against baselines including Temperature Sampling [12], Arithmetic Sampling [16], Diverse Beam Search [14], and Determinantal Beam Search [21] on diverse generation tasks.

## 4.1 Best-of-N Evaluation

### 4.1.1 Experimental Setup

**Datasets.** We evaluate the effectiveness of SemDiD in Best-of-N settings across diverse tasks: *Reasoning tasks* include ARC-Challenge [27], Big Bench Hard (BBH) [28], GSM8K [29], and Minerva Math [30]. *Question answering tasks* include CoQA [31], PubMedQA [32], and MMLU-Pro+ [33]. *Machine translation tasks* include WMT16 [34] (English-German, German-English)

**Evaluation Metrics.** For each query, we generate $N$ responses using different decoding strategies. We measure performance using two key metrics. *Coverage* represents the percentage of test examples with at least one correct answer among the $N$ generated responses. *Accuracy* indicates the percentage of test examples where the selected answer (via majority voting or LLM judge) is correct.

### 4.1.2 Best-of-N Results

Figure 7 demonstrates that SemDiD consistently outperforms other decoding methods across diverse benchmarks. On reasoning tasks, SemDiD shows superior coverage even with fewer samples, achieving 82.4% on ARC-Challenge, 85.6% on BBH, 98.1% on GSM8K, and 86.1% on Minerva Math with just 25 samples. For question answering tasks, SemDiD maintains its advantage with 46.7% on CoQA, 82.6% on PubMedQA, and 82.63% on MMLU-Pro+ at 25 samples. The translation tasks show similar trends with SemDiD reaching 37.2% and 44.7% coverage on WMT16 English-German and German-English respectively at 50 samples. Group Beam Search generally performs well as the second-best method in many tasks, particularly in translation tasks where it closely trails SemDiD, indicating that structured diversity approaches outperform independent sampling methods. Temperature sampling shows inconsistent performance across tasks, with T=1.5 performing relatively well on BBH (77.46% at 25 samples) but underperforming on Minerva Math (71.2% at 25

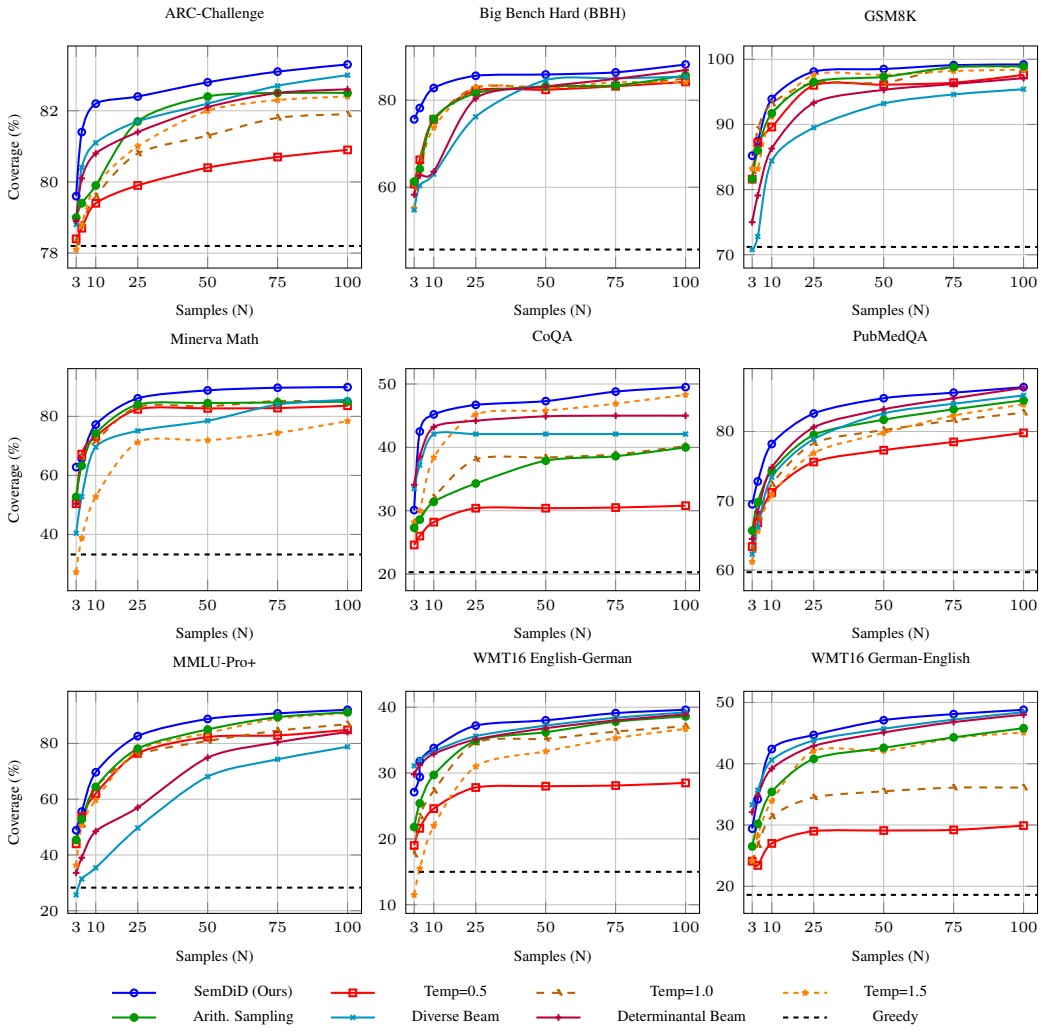

Figure 7: Coverage comparison for Best-of-N using Qwen-2.5-3B.

samples). Notably, all diverse decoding strategies substantially outperform greedy decoding (shown by the horizontal dashed line), with coverage gaps of up to 40 percentage points on certain tasks, underscoring the critical importance of diverse sampling for complex reasoning and generation tasks.

## 4.2 Impact on RLHF Training

### 4.2.1 Experimental Setup

We evaluate SemDiD's impact on reinforcement learning from human feedback (RLHF) by:

*Iterative-RLHF* [5]: Progressively enhances policy models through bootstrapped reward signals in an online learning fashion, utilizing generated responses as training examples.

*DeepSeek-GRPO* [7]: A group-based RLHF algorithm that improves mathematical reasoning by selecting optimal solutions from answer groups.

*RLOO* [6]: A simplified REINFORCE-style optimization that outperforms both PPO and "RL-free" methods with lower computational costs while preserving alignment performance.

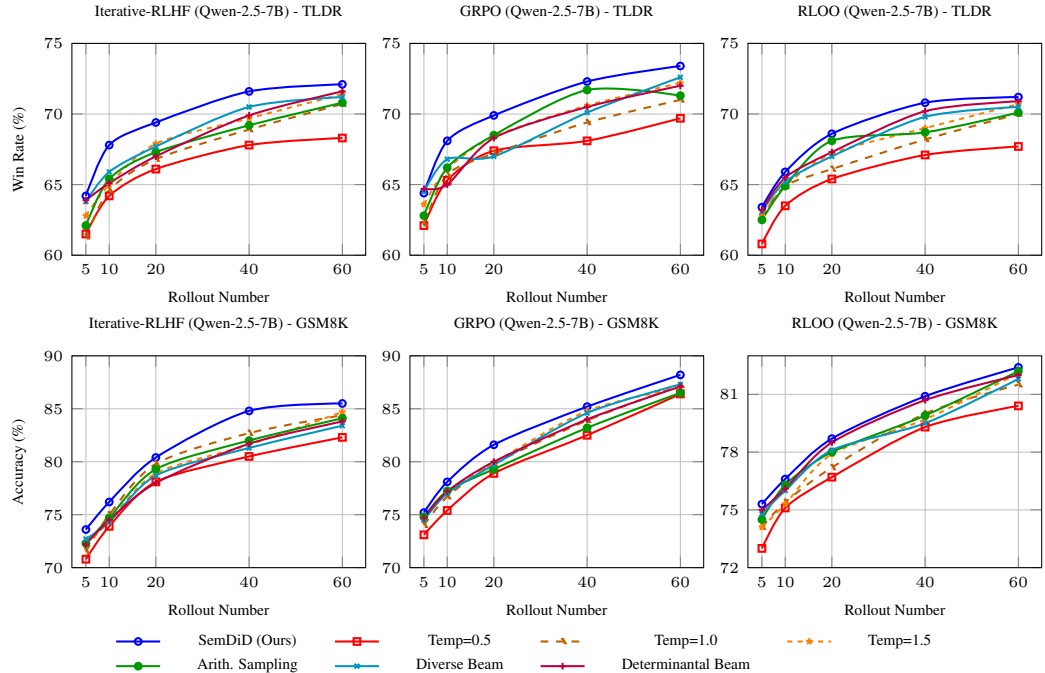

Figure 8: Performance comparison across RLHF algorithms with varying rollout numbers using Qwen-2.5-7B.

For our experiments, we employ Qwen-2.5-7B and Pythia-1B as base models, training them on the mathematical dataset GSM8K[4] and the summarization dataset TLDR[5]. We evaluate performance using accuracy for GSM8K and win rate for TLDR, where the win rate is assessed by GPT-o1-mini against the untrained baseline. For training feedback, we utilize a rule-based reward[6] for GSM8K and a reward model[7] for TLDR.

### 4.2.2 RLHF Results

Figure 8 reveals that SemDiD consistently enhances RLHF training across all evaluated algorithms and tasks. For TLDR summarization, SemDiD achieves the highest win rates of 72.1%, 73.4%, and 71.2% with Iterative-RLHF, GRPO, and RLOO respectively at 60 rollouts. On GSM8K, SemDiD reaches impressive accuracy scores of 85.5% with Iterative-RLHF, 88.2% with GRPO, and 82.4% with RLOO. The performance gap between SemDiD and other methods widens as rollout numbers increase, with minimal differences at 5 rollouts but substantial advantages at 50 rollouts, suggesting that semantic diversity becomes increasingly valuable with extended exploration. Importantly, SemDiD's ability to generate semantically diverse sequences results in more varied reward signals (neither all 0s nor all 1s), creating larger advantage estimates that prevent policy collapse and drive more effective learning. Group-based diversity strategies (Diverse Beam Search and Determinantal Beam Search) consistently outperform independent sampling methods (Temperature Sampling and Arithmetic Sampling), indicating that explicit inter-sequence interactions are crucial for effective exploration in RLHF, with GRPO combined with SemDiD yielding the strongest overall performance.

## 5 Conclusion

We introduced Semantic-guided Diverse Decoding (SemDiD), addressing the critical limitation of semantic diversity in LLM outputs. By operating directly in embedding space through orthogonal

---

[4]https://huggingface.co/datasets/openai/gsm8k

[5]https://huggingface.co/datasets/trl-lib/tldr

[6]https://github.com/huggingface/Math-Verify

[7]https://huggingface.co/trl-lib/pythia-1b-deduped-tldr-rm

directional guidance, dynamic inter-group repulsion, and debiased probability assessment, SemDiD significantly outperforms existing methods. Our approach balances quality and diversity using an $\epsilon$-constraint mechanism and harmonic gain function that adaptively prioritizes the weaker aspect of each candidate. With stage-aware transitions between exploration mechanisms, SemDiD maintains semantic differentiation throughout the generation process. Our extensive evaluations across Best-of-N and RLHF frameworks demonstrate SemDiD's effectiveness in improving response diversity and quality simultaneously without additional training.

## Acknowledgments

We would like to specially thank the support from the B2 project of the HKUST & MetaX Joint Laboratory. The research work described in this paper was supported by Hong Kong Research Grants Council (grant# 16202722, 16210625, T43-513/23-N, T22-607/24N). It was partially conducted in JC STEM Lab of Data Science Foundations funded by The Hong Kong Jockey Club Charities Trust. We acknowledge the support of the National Natural Science Foundation of China (Grant No.62272334, 6257073827, 62577050) and the Natural Science Foundation of Zhejiang Province (Grant No.LY23F020010).

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

# A  Analysis of Probability-Based Quality Assessment

## A.1  Experimental Setup

To investigate the relationship between token probabilities and answer quality, we conducted a series of experiments using mathematical reasoning and common-sense question answering tasks. We sampled from three datasets: GSM8K [29], ARC-Challenge [27], and BBH [28], selecting 500 problems from each. For each problem, we generated 100 candidate answers using nucleus sampling with $p = 0.95$ and temperature $T = 1.0$ by Qwen-2.5-3B. Each generated answer was evaluated using two methods:

**LLM-as-Judge**: A large language model evaluated each answer for correctness (binary classification).

**Probability Thresholding**: Answers were classified as correct/incorrect based on various log probability thresholds.

For LLM-as-Judge, we used a stronger model than the one generating the answers to ensure reliable evaluation, including Qwen-2.5-3B, 7B, and 70B.

## A.2  Probability-Quality Relationship

Our analysis revealed a non-linear relationship between sequence probability and answer quality. As shown in Figure 2, there exists a critical threshold region where quality assessment accuracy changes dramatically.

For log probability thresholds below $-2.5$, accuracy of quality assessment increases rapidly with higher threshold values. This suggests that extremely low-probability sequences strongly correlate with incorrect answers. However, accuracy plateaus in the region between $-2.5$ and $-1.0$, indicating diminishing returns for higher probability thresholds. When log probability thresholds exceed $-1.0$, we observed a significant decrease in assessment accuracy. This unexpected decline occurs because many correct answers were misclassified as incorrect at these higher thresholds. The finding suggests that probability is better suited as a lower bound to filter out low-quality outputs rather than as a maximization target.

Additionally, we observed that model size significantly impacts the effectiveness of the LLM-as-judge approach. The 70B model achieved an accuracy of 0.8185. While smaller 7B and 3B models reached only 0.7685 and 0.7417 respectively, comparable to probability thresholding. However, larger models inevitably introduce substantial inference costs, making them impractical for real-time evaluation during the decoding process.

## A.3  Position and Length Bias Analysis

Further analysis revealed systematic biases in token probabilities that compromise fair quality assessment:

### A.3.1  Position Bias

We computed average log probabilities for tokens at different positions across all generated sequences. As shown in Figure 3, a clear upward trend emerges: tokens appearing later in sequences receive consistently higher probability scores. This effect is particularly pronounced after position 20, where average log probabilities increase by approximately 0.15 per 10 tokens.

This position bias creates an unfair advantage for longer responses in probability-based quality assessment, as their aggregate scores benefit from the artificially inflated probabilities of later tokens.

**Initial Token Bias.** We also observed significantly higher confidence (i.e., higher probability) for initial tokens across generated responses. This phenomenon occurs because language models are trained on common opening phrases such as "To solve this problem...", "According to the given information...", or "Let's approach this step by step...".

### A.3.2 Sentence Progress Bias

Similar patterns appear within sentence structures. We analyzed the relationship between a token's distance from the most recent sentence-ending punctuation and its assigned probability. Figure 4 illustrates that tokens appearing further into a sentence receive progressively higher probabilities, with an average increase of 0.08 per 5 tokens after punctuation. This sentence-level bias favors certain syntactic structures and further distorts quality assessment when using raw probability scores.

### A.4 Debiasing Methodology Evaluation

To address these biases, we implemented the position-debiased probability described in Section 3.1. We conducted an ablation study to evaluate the effectiveness of our debiasing approach on SemDiD, which used the Qwen-2.5-3B model with 10 samples per question. The combined approach with saturation threshold achieved the highest accuracy, validating our debiasing methodology.

Table 1: Performance comparison of different debiasing approaches under 10 samples per question.

| Method | Avg. Coverage | Avg. Accuracy by LLM-as-Judge |
|---|---|---|
| Raw Probability | 67.85% | 61.02% |
| Position Debiasing Only | 68.41% | 61.90% |
| Sentence Debiasing Only | 67.95% | 61.23% |
| Combined Debiasing | 68.64 % | 62.36% |
| + Saturation Threshold | 68.76 % | 62.47% |

## B  Analysis of Semantic Diversity Assessment

### B.1 Experimental Setup

To investigate the value of semantic diversity in response generation, we conducted experiments comparing different selection strategies for Best-of-N evaluation. Using GSM8K and BBH reasoning problems as our testbed, we generated 100 candidate responses for each problem using nucleus sampling with $p = 0.95$ and temperature $T = 1.0$ with Qwen-2.5-3B. We then applied various selection strategies to identify the most representative diverse subsets.

### B.2 Selection Strategies

We compared three distinct selection methods:

**Highest Probability**: The standard approach of selecting answers based solely on their log probability scores, taking the top-k candidates with highest probabilities.

**N-gram Clustering**: Candidates were clustered based on character-level n-gram similarity. For each cluster, we selected the representative with the highest probability score. This approach ensures lexical diversity but may not capture deeper semantic differences.

**Semantic Clustering**: We embedded all 100 candidates using 0.5B and 1.5B parameter sentence embedding models, performed K-means clustering in this semantic space, and selected the highest-probability candidate from each cluster.

### B.3 Semantic Space Visualization

Figure 5 provides a t-SNE visualization of the semantic embedding space for a specific GSM8K problem, where correct answers (circles) and incorrect answers (crosses) are scattered across distinct semantic clusters. The problem analyzed is:

> **GSM8K Problem Example**
>
> A rectangular band formation is a formation with $m$ band members in each of $r$ rows, where $m$ and $r$ are integers. A particular band has less than 100 band members. The director arranges them in a rectangular formation and finds that he has two members left over. If he increases the number of members in each row by 1 and reduces the number of rows by 2, there exactly enough places in the new formation for each band member. What is the largest number of members the band could have?

For this problem, 100 responses were generated, of which 22 were correct answers. These responses were clustered into 10 semantic groups. Notably, 9 of the 10 clusters contained at least one correct answer, and in 5 clusters, the highest-probability response was correct. Several key observations emerge:

**Multiple Valid Solution Paths**: Correct answers appear in multiple distinct clusters rather than concentrated in a single region, confirming that multiple valid reasoning paths exist for solving mathematical problems.

**Probability-Correctness Correlation**: Circle sizes represent probability magnitudes, showing that while higher probability generally correlates with correctness within clusters, this relationship is not uniform across the entire semantic space.

**Cluster Purity**: Some semantic clusters contain predominantly correct answers (e.g., purple, blue, and orange clusters), while others show a mix of correct and incorrect solutions (e.g., red and yellow clusters). This suggests that certain reasoning approaches are inherently more reliable than others.

**Distribution Patterns**: Incorrect answers also form clusters, indicating systematic error patterns that could potentially be useful for model improvement and error analysis.

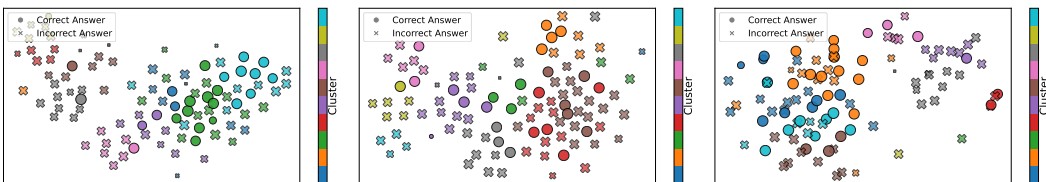

Figure 9: t-SNE visualization of semantic embedding space for GSM8K problems, where correct answers (circles) and incorrect answers (crosses) are scattered across distinct semantic clusters. Circle sizes represent probability magnitudes.

More examples are provided in Figure 9. This visualization highlights the importance of exploring different semantic regions during decoding rather than simply maximizing token probabilities, as valuable correct answers may exist across diverse semantic clusters rather than being concentrated in a single high-probability region.

## B.4 Results and Analysis

As shown in Figure 6, semantic clustering consistently outperforms both probability-based ranking and character-level approaches across all sample sizes. When selecting top-k diverse candidates from 100 samples per problem, semantic clustering with even the smaller 0.5B model achieves up to 95% coverage at k=25, compared to 92% for regular probability-based selection. The performance gap is particularly pronounced at lower sample counts (k=3,5,7), where resource efficiency is most critical. At k=3, semantic clustering provides a 3-4% absolute improvement in coverage compared to highest probability selection.

**Embedding Model Size Impact**: Surprisingly, we found minimal difference between the 0.5B and 1.5B embedding models in terms of clustering effectiveness. The 0.5B model achieved 91.7% coverage at k=10, only 0.1% below the 1.5B model's 91.8%. This finding suggests that even lightweight embedding models can effectively capture the semantic distinctions necessary for diversity assessment, allowing for computational efficiency without sacrificing diversity quality.

**Implications for Diverse Decoding.** Our analysis demonstrates that semantic diversity provides significant advantages over lexical diversity or probability-based selection in Best-of-N scenarios. The findings support the design decisions in SemDiD, particularly the use of directional guidance to explore distinct semantic regions rather than relying solely on token-level diversity, the importance of balancing quality (via probability) with semantic diversity, and the viability of using lightweight embedding models for semantic assessment during the decoding process. Furthermore, these results validate the efficacy of semantic clustering as a post-hoc selection strategy when multiple candidate responses are available. For practical applications where generating a large number of candidates is feasible, semantic clustering provides an effective mechanism for identifying a diverse and representative subset.

## C Algorithm and Optimization Details

### C.1 SemDiD Algorithm

Algorithm 1 provides a detailed overview of the Semantic-guided Diverse Decoding (SemDiD) procedure. The algorithm simultaneously manages $k$ groups of beam searches, with the first group following greedy decoding to establish a quality baseline.

### C.2 Efficient Implementation

A naive implementation of the SemDiD algorithm would incur substantial computational overhead due to duplicated embedding calculations. To address this, we implemented several optimization techniques:

#### C.2.1 Dynamic Multi-step Lookahead

Unlike traditional Diverse Beam Search methods that rely on lightweight heuristic rules for token selection, operating in the semantic space requires embedding computations that could become prohibitively expensive. Traditional approaches evaluate all tokens in the vocabulary (typically around 100K tokens) against beam scoring functions, which is computationally feasible for simple n-gram penalties.

With SemDiD, evaluating the semantic impact of each vocabulary token through embedding calculations is impractical. Moreover, the semantic change from a single token is often too subtle for reliable diversity assessment. To address this, we implement dynamic multi-step lookahead that explores $E_t$ sequences per beam, each extending forward by $L_t$ steps. It allows us to evaluate more substantial semantic deviations rather than token-level variations.

The exploration width $E_t$ is dynamically adjusted based on sentence position [35, 36]:

$$E_t = \begin{cases} E_{base} \cdot \mu_p, & \text{if position } t \text{ follows punctuation} \\ E_{base} \cdot \mu_n, & \text{otherwise} \end{cases} \tag{15}$$

where $\mu_p$ and $\mu_n$ are hyperparameters, set to 1.5 and 0.7 respectively in our implementation. Setting smaller values helps reduce the computational cost of SemDiD. Alternatively, a linear function of sentence position can also be used to adjust these parameters dynamically.

We set a maximum lookahead depth of $L_{max}$, but terminate the exploration early if a punctuation mark is encountered:

$$\text{ExploreDepth}(y) = \min(L_{max}, \text{StepsUntilNextPunctuation}(y)) \tag{16}$$

This adaptive approach ensures that we explore complete semantic units (sentences or clauses) without wasting computational resources on unnecessarily deep lookahead. Our analysis shows that approximately 26% of decoding steps qualify for increased exploration width. While we considered using entropy-based dynamic adjustments, token-level entropy exhibits high variance and can introduce noise, so we opted for the simpler punctuation-based approach.

**Algorithm 1** Semantic-guided Diverse Decoding (SemDiD)

---

**Require:** Query $q$, language model $M$, embedding model $\mathcal{E}$, number of groups $k$, beam size $b$, quality relaxation $\gamma$, transition step $T_{trans}$, harmonic strength $\lambda$

**Ensure:** $k$ semantically diverse responses $\{y_1, y_2, \ldots, y_k\}$

1: Initialize $Y_0^1 \leftarrow \{q\}$          $\triangleright$ Greedy group has single beam
2: Initialize $Y_0^g \leftarrow \{q\}$ for groups $g \in [2, k]$          $\triangleright$ Diverse groups start with query
3: Initialize $\vec{r}_g \leftarrow \text{RandomUnitVector}()$ for all groups $g \in [1, k]$          $\triangleright$ Random vectors
4: Initialize group directions $\vec{d}_g \leftarrow$ null for all groups $g \in [1, k]$
5: **for** $t = 1, 2, \ldots, T_{\max}$ **do**
6:      $\alpha_t \leftarrow \min(1, \frac{t}{T_{trans}})$          $\triangleright$ Update transition weight
7:      **for** $g = 1, 2, \ldots, k$ **do**          $\triangleright$ Process each group in parallel
8:          $C_t^g \leftarrow \emptyset$          $\triangleright$ Candidates for group $g$ at step $t$
9:          **if** $t \bmod T_{update} = 1$ **then**          $\triangleright$ Update directions periodically
10:              **if** $g = 1$ **then**
11:                  Let $y^1$ be the sequence in group 1
12:                  $\vec{d}_1 \leftarrow \frac{\mathcal{E}(y^1) - \mathcal{E}(q)}{||\mathcal{E}(y^1) - \mathcal{E}(q)||}$          $\triangleright$ First group direction
13:              **else**
14:                  $\vec{v}_g \leftarrow \vec{r}_g - \sum_{i=1}^{g-1} \frac{\vec{r}_g \cdot \vec{d}_i}{||\vec{d}_i||^2} \vec{d}_i$          $\triangleright$ Gram-Schmidt
15:                  $\vec{d}_g \leftarrow \frac{\vec{v}_g}{||\vec{v}_g||}$          $\triangleright$ Normalize direction
16:              **end if**
17:          **end if**
18:          **if** $g = 1$ **then**          $\triangleright$ Greedy group with single beam
19:              Let $y$ be the sequence in $Y_{t-1}^1$
20:              $y' \leftarrow \text{GreedyLookahead}(y, L_t)$          $\triangleright$ Greedy lookahead
21:              $S_{\text{quality}}(y') \leftarrow \text{DebiasedProb}(p_M(y'|y), |y'| - |y|, d_{punct})$
22:              $Y_t^1 \leftarrow \{y'\}$          $\triangleright$ Maintain single beam
23:              Update $\epsilon \leftarrow \min(\epsilon, S_{\text{quality}}(y') \cdot \gamma)$          $\triangleright$ Update threshold
24:          **else**          $\triangleright$ Diverse groups with multiple beams
25:              **for** $y \in Y_{t-1}^g$ **do**          $\triangleright$ For each beam in the group
26:                  $E_t \leftarrow \text{DetermineExplorationWidth}(y)$          $\triangleright$ Dynamic exploration width
27:                  **for** $e = 1, 2, \ldots, E_t$ **do**          $\triangleright$ Generate $E_t$ lookahead sequences
28:                      $y' \leftarrow \text{GenerateLookahead}(y, L_t)$          $\triangleright$ Generate one path
29:                      $S_{\text{quality}}(y') \leftarrow \text{DebiasedProb}(p_M(y'|y), |y'| - |y|, d_{punct})$
30:                      $S_{\text{dir}}(y') \leftarrow \cos\left(\mathcal{E}(y') - \mathcal{E}(q), \vec{d}_g\right)$          $\triangleright$ Directional score
31:                      $S_{\text{rep}}(y') \leftarrow -\max_{g' \neq g} \langle \mathcal{E}(y'), \mathcal{E}(\text{Best}(Y_{t-1}^{g'})) \rangle$          $\triangleright$ Repulsion
32:                      $C_t^g \leftarrow C_t^g \cup \{(y', S_{\text{quality}}(y'), S_{\text{dir}}(y'), S_{\text{rep}}(y'))\}$
33:                  **end for**
34:              **end for**
35:              Use percentile-based normalization for $S_{\text{quality}} \rightarrow \tilde{S}_{\text{quality}}, S_{\text{dir}} \rightarrow \tilde{S}_{\text{dir}}, S_{\text{rep}} \rightarrow \tilde{S}_{\text{rep}}$
36:              **for** each candidate $(y', *, *, *)$ in $C_t^g$ **do**
37:                  $\tilde{S}_{\text{div}}(y') \leftarrow (1 - \alpha_t) \cdot \tilde{S}_{\text{dir}}(y') + \alpha_t \cdot \tilde{S}_{\text{rep}}(y')$
38:                  **if** $S_{\text{quality}}(y') < \epsilon$ **then**
39:                      $S_{\text{combined}}(y') \leftarrow -\infty$          $\triangleright$ Below quality threshold
40:                  **else**
41:                      $S'_{\text{quality}}(y') \leftarrow \max(0, \tilde{S}_{\text{quality}}(y') - \epsilon)$          $\triangleright$ Quality surplus
42:                      $S_{\text{combined}}(y') \leftarrow \frac{\lambda \cdot S'_{\text{quality}}(y') \cdot \tilde{S}_{\text{div}}(y')}{S'_{\text{quality}}(y') + \tilde{S}_{\text{div}}(y')}$          $\triangleright$ Harmonic
43:                  **end if**
44:              **end for**
45:              $Y_t^g \leftarrow \text{SelectTopK}(C_t^g, b)$          $\triangleright$ Keep top-b candidates per group
46:          **end if**
47:      **end for**
48:      **if** all groups have complete responses or reached max length **then**
49:          **break**
50:      **end if**
51: **end for**
52: **return** $\{Y_T^1, \text{Best}(Y_T^2), \ldots, \text{Best}(Y_T^k)\}$          $\triangleright$ Return best from each diverse group

---

Each group functions as an independent beam search representing a distinct search direction. The groups explore asynchronously to reduce computational bottlenecks. After every $T_{update}$ steps of greedy decoding, we update the reference direction and guidance vectors for each group.

### C.2.2 KV-Cache Utilization for Embedding Efficiency

To further reduce computational overhead, we leverage the key-value (KV) cache mechanism when computing embeddings. We recommend using autoregressive embedding models such as Salesforce/SFR-Embedding-2_R [37] or NovaSearch/stella_en_1.5B_v5 [38] (we chose in this paper), as they allow for efficient reuse of computation in sequential token processing. These models can share the same KV cache architecture used in language model decoding.

Alternatively, small BERT-style models like the 30M all-MiniLM-L6-v2 [8] can be used without introducing excessive computational costs. Our experiments indicate that embedding model size has minimal impact on diversity quality, making lightweight models a practical choice for production deployments.

### C.3 Computational Complexity Analysis

**Theorem 4 (Computational Complexity)** *The time complexity of SemDiD is:*

$$O(k \cdot b \cdot T \cdot E_t \cdot (L_t \cdot C_{LM} + C_E)) \tag{17}$$

*where $k$ is the number of groups, $b$ is the beam size, $T$ is the total decoding steps, $E_t$ is the exploration width per beam, $L_t$ is the lookahead depth, $C_{LM}$ is the cost of a language model forward pass, and $C_E$ is the cost of an embedding model forward pass.*

**Proof:** SemDiD maintains $k$ parallel groups executing beam search until sequence completion, requiring $T$ total decoding steps. The first group performs greedy decoding with a single beam ($b = 1$), while groups 2 through $k$ maintain $b$ beams each.

At each decoding step $t$, the greedy group generates one lookahead sequence extending $L_t$ tokens from its current state. This requires $L_t$ sequential forward passes through the language model, resulting in $L_t \cdot C_{LM}$ computational cost per step.

For diverse groups (2 through $k$), each beam explores $E_t$ different continuation paths at every step. Each continuation generates a lookahead sequence of length $L_t$, requiring $L_t$ language model forward passes. After generating each complete lookahead sequence, the algorithm computes its embedding for semantic diversity assessment. Therefore, each beam in diverse groups incurs $E_t \cdot (L_t \cdot C_{LM} + C_E)$ cost per step.

The total computational cost across all $T$ steps is:

$$\text{Cost} = T \cdot [\text{Greedy Group} + \text{Diverse Groups}] \tag{18}$$
$$= T \cdot [L_t \cdot C_{LM} + (k - 1) \cdot b \cdot E_t \cdot (L_t \cdot C_{LM} + C_E)] \tag{19}$$

Since the greedy group can be viewed as having $E_t = 1$ and $b = 1$, and noting that it doesn't require embedding computation, this simplifies to the stated complexity $O(k \cdot b \cdot T \cdot E_t \cdot (L_t \cdot C_{LM} + C_E))$.

KV-cache optimization significantly reduces the effective cost of forward passes by reusing previously computed key-value states. Early termination at punctuation marks bounds the actual lookahead depth below $L_{max}$, while dynamic adjustment of $E_t$ based on sentence position further optimizes resource utilization.

Traditional Diverse Beam Search operates with complexity $O(k \cdot b \cdot T \cdot (L_t \cdot C_{LM} + V \cdot C_{score}))$, where $V$ is the vocabulary size and $C_{score}$ is the cost of evaluating diversity penalties. At each step, DBS performs a forward pass through the language model to obtain token probabilities, then evaluates diversity penalties across all $V$ vocabulary tokens. While $C_{score} \ll C_{LM}$, scoring 100K tokens creates small overhead. The key distinction is that DBS applies shallow, token-level diversity penalties, whereas SemDiD invests computation in deeper semantic exploration through multi-token lookahead.

---

[8] https://huggingface.co/sentence-transformers/all-MiniLM-L6-v2

Temperature sampling achieves $O(k \cdot T \cdot L_t \cdot C_{LM})$ complexity per sequence, appearing more efficient. However, achieving comparable semantic diversity requires generating more independent samples where $k_{temp} \gg k_{semDiD} \cdot b$, resulting in total complexity that may exceed SemDiD's cost.

In practice, despite introducing threefold computational requirements, the actual end-to-end latency of SemDiD increases by only 25-35% compared to group beam search (huggingface official implementation) with typical parameters ($k = 25$, $b = 3$, $E_t \approx 3$, $L_t \approx 10$), making it a practical solution for real-world applications. This modest increase in time cost delivers substantial improvements in semantic diversity and downstream performance, creating a favorable trade-off between resource consumption and quality gains for applications where diverse, high-quality outputs are critical. Under conditions of unlimited computational resources or for high-value applications such as RLHF training, the significant performance improvements that SemDiD delivers greatly outweigh its marginal computational costs.

# D Hyperparameter Settings and Analysis

## D.1 Default Hyperparameter Settings

Table 2 presents the default hyperparameters used in our experiments. These values were determined through extensive grid search optimization on a held-out validation set. All experiments were conducted on a cluster of 8 NVIDIA H800 GPUs. The Best-of-N evaluations across all benchmarks required approximately 70 hours, while each RLHF training took approximately 40 hours for Qwen-2.5-3B.

Table 2: Default hyperparameter settings for SemDiD

| Parameter | Description | Default Value |
|---|---|---|
| $N$ | Number of groups | **k** or k/2 |
| $b$ | Beam size per group | **5** / 3 |
| $E_{base}$ | Base exploration width | **4** / 3 |
| $L_{max}$ | Maximum lookahead depth | 20 / **10** / 5 |
| $T_{trans}$ | Transition step for $\alpha_t$ | 10 |
| $\gamma$ | Quality relaxation parameter | 0.25 |
| $\lambda$ | Harmonic gain strength | 2.0 |
| $\beta_{seq}$ | Sequence position decay rate | 0.001 |
| $\beta_{sent}$ | Sentence position decay rate | 0.005 |
| $\tau$ | Probability saturation threshold | -0.8 |
| Temperature | Sampling temperature | 1.0 |
| Top-p | Top-p (nucleus sampling) threshold | 0.95 |

The number of groups $N$ is adaptively set to either the target number of samples $k$ or $k/2$, depending on the desired trade-off between computational cost and diversity quality. When $N = k$, each group returns one candidate, maximizing inter-group diversity. When $N = k/2$, each group returns two candidates, reducing computational overhead while maintaining reasonable diversity.

## D.2 Exploration Width and Beam Size Analysis

We conducted experiments to analyze the relationship between exploration width ($E_t$) and beam size ($b$) on the GSM8K dataset. Figure 10 shows the coverage achieved with different parameter combinations.

Results show that increasing the exploration width from 1 to 4 yields significant improvements in coverage. For instance, with $b = 3$, coverage improves from 87.3% to 93.8%, representing a 6.5 percentage point gain. However, further increases to $E_t = 8$ and $E_t = 16$ provide diminishing returns, with coverage plateauing at around 94.7%. This saturation effect can be attributed to the limited semantic variance available in most problems, where exploring beyond a certain width captures only redundant variations.

Similarly, beam size shows a positive correlation with coverage up to $b = 3$, after which the gains become marginal. For example, at $E_t = 4$, increasing beam size from $b = 1$ to $b = 3$

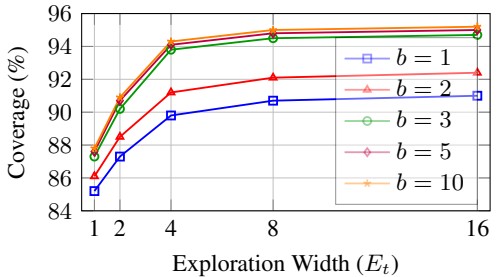

Figure 10: Impact of exploration width and beam size on coverage. Results are averaged over 500 GSM8K problems with 10 groups.

improves coverage from 89.8% to 93.8%, but further increasing to $b = 10$ only yields an additional 0.5 percentage points. The computational cost, however, increases linearly with both $E_t$ and $b$, suggesting that $E_t = 4$ and $b = 3$ offer the optimal trade-off between performance and efficiency.

### D.3 Group Structure Analysis

We investigated an alternative group configuration where each group returns multiple candidates rather than a single representative. Table 3 presents the coverage results when varying the number of outputs per group while maintaining a constant total of 10 outputs.

Table 3: Coverage comparison with different group structures on GSM8K. Total outputs fixed at 10.

| Outputs per Group | Number of Groups | Coverage (%) |
| --- | --- | --- |
| 1 | 10 | 94.8 |
| 2 | 5 | 94.5 |
| 3 | 3 (with 1 group returning 4) | 92.2 |
| 5 | 2 | 87.9 |
| 10 | 1 | 84.3 |

The results demonstrate that the single-output-per-group configuration achieves optimal coverage. When groups return multiple candidates, coverage remains relatively stable for 2-3 outputs per group but degrades significantly beyond that. This pattern suggests that inter-group diversity mechanisms are more effective than intra-group selection for maintaining semantic separation.

The decline in coverage with fewer groups can be explained by the reduced effectiveness of orthogonal direction guidance. With fewer groups, the semantic space cannot be partitioned as efficiently, leading to increased overlap between exploration regions. Additionally, the inter-group repulsion mechanism becomes less effective with fewer groups, as there are fewer distinct trajectories to maintain separation between.

The choice of using multiple groups with single outputs as the default configuration, while allowing flexibility to use $N = k/2$ when computational resources are constrained. The modest decrease in coverage (0.3%) when using two outputs per group makes this an attractive option for resource-limited settings.

### D.4 Lookahead Depth Analysis

Unlike character-level diversity methods that can evaluate each vocabulary token individually with minimal computational cost, semantic diversity assessment requires embedding model forward passes, making token-by-token evaluation prohibitively expensive. Additionally, single-token semantic changes are often too subtle for reliable diversity measurement. Therefore, SemDID introduces the lookahead depth parameter $L_{max}$ to control how many tokens ahead we explore when evaluating semantic diversity, allowing assessment of more substantial semantic deviations while managing computational overhead.

We conducted experiments using Qwen-2.5-3B across different $L_{max}$ values to demonstrate its sensitivity and identify optimal settings. Table 4 shows the performance and computational overhead analysis across different lookahead depths.

Table 4: Performance and computational overhead analysis across $L_{max}$ values using Qwen-2.5-3B.

| $L_{max}$ | GSM8K Coverage (N=25) | ARC Coverage (N=25) | Computational Overhead |
|---|---|---|---|
| 5 | 95.3% | 80.1% | +15% |
| 10 | 98.1% | 82.4% | +25% |
| 15 | 98.2% | 82.6% | +35% |
| 20 | 98.6% | 82.7% | +45% |

Performance saturates around $L_{max} = 10$, with diminishing returns for deeper lookahead while computational overhead continues to increase substantially. This suggests that 10 tokens provide sufficient context for reliable semantic diversity assessment without excessive computational cost.

### D.5 Quality-Diversity Balancing Analysis

The quality relaxation parameter $\gamma$ and harmonic strength $\lambda$ control the trade-off between maintaining quality thresholds and pursuing semantic diversity. We conducted systematic sensitivity analysis by varying each parameter independently to understand their effects on both coverage and accuracy.

**Effect of $\gamma$ (with $\lambda = 2.0$ fixed):** The quality relaxation parameter $\gamma$ determines how much the quality threshold can be relaxed to promote diversity. Lower values maintain stricter quality requirements, while higher values allow more quality sacrifice for diversity gains.

Table 5: Sensitivity analysis of quality relaxation parameter $\gamma$ with $\lambda = 2.0$ fixed.

| Task | $\gamma = 0.15$ | $\gamma = 0.20$ | $\gamma = 0.25$ | $\gamma = 0.30$ | $\gamma = 0.35$ |
|---|---|---|---|---|---|
| GSM8K Coverage (N=25) | 96.6% | 97.1% | 98.1% | 97.8% | 97.4% |
| GSM8K Accuracy (N=25) | 75.9% | 76.6% | 77.5% | 77.2% | 77.2% |
| WMT16 Coverage (N=25) | 36.7% | 36.8% | 37.2% | 36.9% | 36.7% |
| WMT16 Accuracy (N=25) | 20.2% | 20.4% | 20.7% | 20.5% | 20.3% |

**Effect of $\lambda$ (with $\gamma = 0.25$ fixed):** The harmonic strength parameter $\lambda$ controls the intensity of the harmonic gain mechanism that balances quality and diversity objectives. Higher values strengthen the diversity promotion effect, while lower values prioritize quality preservation.

Table 6: Sensitivity analysis of harmonic strength parameter $\lambda$ with $\gamma = 0.25$ fixed.

| Task | $\lambda = 1.0$ | $\lambda = 1.5$ | $\lambda = 2.0$ | $\lambda = 2.5$ | $\lambda = 3.0$ |
|---|---|---|---|---|---|
| GSM8K Coverage (N=25) | 96.8% | 97.4% | 98.1% | 97.9% | 97.6% |
| GSM8K Accuracy (N=25) | 77.3% | 77.1% | 77.5% | 77.3% | 76.9% |
| WMT16 Coverage (N=25) | 36.8% | 36.5% | 37.2% | 36.9% | 36.7% |
| WMT16 Accuracy (N=25) | 20.3% | 20.6% | 20.7% | 20.7% | 20.6% |

The results demonstrate that our default settings ($\gamma = 0.25$, $\lambda = 2.0$) consistently achieve near-optimal performance across different tasks, with performance remaining stable within reasonable parameter ranges. The sensitivity analysis shows that SemDiD is robust to parameter variations, with performance degrading gracefully rather than sharply when moving away from optimal values.

### D.6 Transition Weight Analysis

The parameter $T_{trans}$ controls the transition point from directional guidance to inter-group repulsion. We conducted experiments varying $T_{trans}$ across multiple datasets:

The results show clear optimal performance at $T_{trans} = 10$. When transition occurs too early ($T_{trans} = 5$), groups haven't established sufficient semantic differentiation before repulsion dominates, leading to suboptimal exploration. When transition is delayed ($T_{trans} \geq 20$), groups may converge to similar semantic regions before inter-group repulsion becomes effective.

Table 7: Transition weight parameter $T_{trans}$ analysis across datasets, where Cov. denotes coverage and Acc. denotes accuracy.

| $T_{trans}$ | GSM8K Cov. | GSM8K Acc. | ARC Cov. | ARC Acc. | MMLU-Pro+ Cov. |
|---|---|---|---|---|---|
| 5 | 97.2% | 76.8% | 81.2% | 79.1% | 81.4% |
| 10 (default) | 98.1% | 77.5% | 82.4% | 82.0% | 82.6% |
| 15 | 97.9% | 77.2% | 82.1% | 81.7% | 82.3% |
| 20 | 97.5% | 76.9% | 81.7% | 81.2% | 81.8% |
| 25 | 97.2% | 76.8% | 81.8% | 80.9% | 81.6% |

**Intuitive Explanation:** $T_{trans} = 10$ corresponds to the typical number of tokens needed to establish meaningful semantic context. Most mathematical reasoning problems require 8-12 tokens to establish the early thought process for problem-solving before semantic trajectories become distinguishable.

### D.7 Guidance for Hyperparameter Settings

Due to SemDiD's involvement of numerous hyperparameters, we provide the following guidelines for setting them based on the comprehensive analysis presented above.

Firstly, temperature, Top-p, $N$ (number of groups), and $b$ (beam size) are inherited from standard Group Beam Search parameters, not SemDID additions. The SemDID-specific parameters serve distinct purposes across three categories: semantic diversity assessment ($E_t$, $L_{max}$), quality assessment ($\beta_{seq}$, $\beta_{sent}$, $\tau$), and quality-diversity balancing ($T_{trans}$, $\gamma$, $\lambda$).

**Automatically Derivable Parameters:** Several key parameters can be systematically determined rather than manually tuned. The position bias parameters $\beta_{seq}$ and $\beta_{sent}$ can be automatically fitted using `scipy.curve_fit` from probability-position curves shown in Figures 3 and 4, as these patterns remain consistent across tasks. The saturation threshold $\tau$ is derived from probability-quality analysis as Figure 2 and set to -0.8 for most tasks based on the empirical study. As demonstrated in Section D.6, the transition point $T_{trans} = 10$ corresponds to the typical number of tokens needed to establish meaningful semantic context before inter-group repulsion becomes effective.

**Resource-Dependent Parameters:** The exploration parameters $E_t$, $b$, and $N$ balance exploration breadth versus computational cost. Our analysis in Sections D.2 and D.3 shows diminishing returns beyond $E_t = 4$ and $b = 3$, establishing clear "sweet spots" without extensive tuning requirements. The lookahead depth analysis in Section D.4 demonstrates that $L_{max} = 10$ provides optimal performance-cost balance.

**Quality-Diversity Balance:** The quality-diversity balancing analysis in Section D.5 reveals that our default settings ($\gamma = 0.25$, $\lambda = 2.0$) achieve robust performance across tasks, with graceful degradation when parameters deviate from optimal values. The transition weight analysis in Section D.6 confirms that $T_{trans} = 10$ provides optimal timing for the shift from directional guidance to inter-group repulsion.

For practitioners seeking to deploy SemDiD, we recommend starting with our provided default parameters for initial implementation, adjusting $E_t$, $b$, and $N$ based on available computational budget, and fine-tuning $\gamma$, $\lambda$, and $T_{trans}$ only for highly specialized applications where task-specific optimization is critical.

## E   Accuracy Evaluation in Best-of-N

To provide a comprehensive evaluation of our SemDiD approach, we conducted experiments measuring the accuracy of the selected answers under Best-of-N settings. Unlike coverage, which measures whether at least one correct answer exists among the N candidates, accuracy evaluates whether we can effectively identify the best answer from the generated candidates. For this purpose, we employed LLM-Blender PairRM [9] as our evaluation model.

For each query, we generated N responses using Qwen-2.5-3B. We then used the LLM-Blender PairRM model to evaluate and select the most likely correct answer from among the candidates. This

---

[9]https://huggingface.co/llm-blender/PairRM

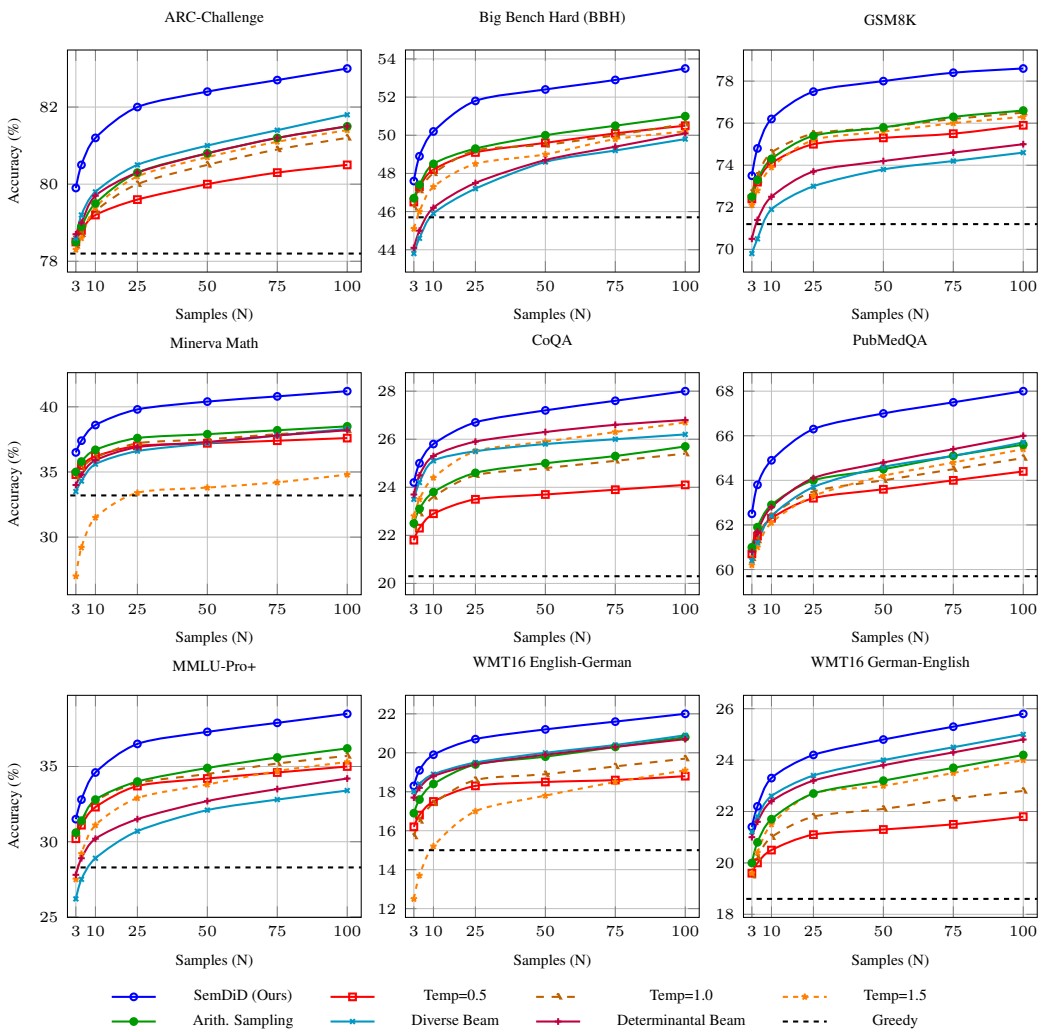

Figure 11: Accuracy comparison for Best-of-N using Qwen-2.5-3B with LLM-Blender PairRM as the evaluation model.

process simulates a real-world scenario where a verification mechanism must determine the optimal answer from a set of alternatives.

Figure 11 presents accuracy results across all benchmarks. The results demonstrate that SemDiD consistently outperforms baseline methods across all tasks, though by modest margins. This indicates that while various decoding strategies can increase the likelihood of generating at least one correct answer (coverage), the ability to identify the best answer (accuracy) remains challenging. Interestingly, we observe that performance differences between methods become more pronounced as the number of samples increases, suggesting that semantic diversity becomes increasingly valuable with larger candidate pools. Mathematical reasoning tasks (GSM8K and Minerva Math) show the greatest improvements with SemDiD, likely because these tasks benefit most from exploring diverse solution paths. Translation tasks exhibit the smallest performance gaps, potentially due to their more constrained solution spaces compared to open-ended reasoning tasks.

# F  Cross-Model Generalization Analysis

In addition to our primary experiments with Qwen-2.5-3B for Best-of-N and Qwen-2.5-7B for RLHF, we extended our analysis to different model architectures and sizes to assess how SemDiD's

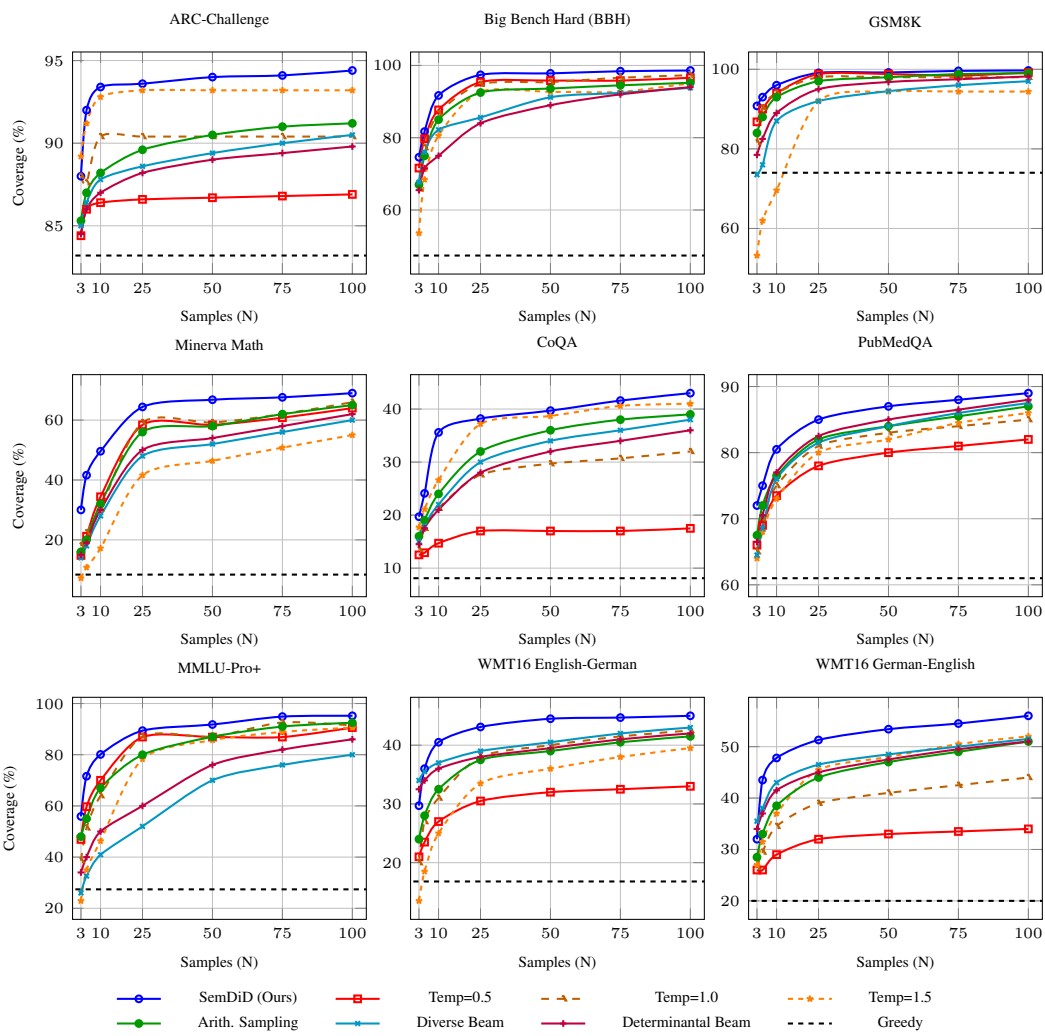

Figure 12: Coverage comparison for Best-of-N using Llama-3.1-8B.

effectiveness generalizes across different language models. Specifically, we evaluated SemDiD using Llama-3.1-8B for Best-of-N tasks and Pythia-1B for RLHF training.

### F.1 Llama-3.1-8B in Best-of-N

For the Best-of-N experiments with Llama-3.1-8B, we followed the same experimental setup as with Qwen-2.5-3B, evaluating performance across all nine benchmarks. As shown in Figure 12, SemDiD consistently outperforms baseline methods across all tasks with Llama-3.1-8B, demonstrating even more pronounced improvements compared to Qwen-2.5-3B. The performance gains are particularly striking on reasoning tasks, with SemDiD achieving 99.1% coverage on GSM8K and 97.4% coverage on BBH at just 25 samples - improvements of 0.5-1.8% over other methods. For language understanding tasks, SemDiD shows exceptional performance on MMLU-Pro+ with 89.4% coverage at 25 samples versus 86.9% for Temperature sampling (T=0.5). On translation tasks, while Diverse Beam Search remains competitive, SemDiD maintains a clear advantage with 43.1% and 51.3% coverage on WMT16 English-German and German-English respectively at 25 samples. These results suggest that SemDiD's semantic diversity mechanisms are particularly effective when combined with larger, more capable models, further validating the architecture-agnostic nature of our approach.

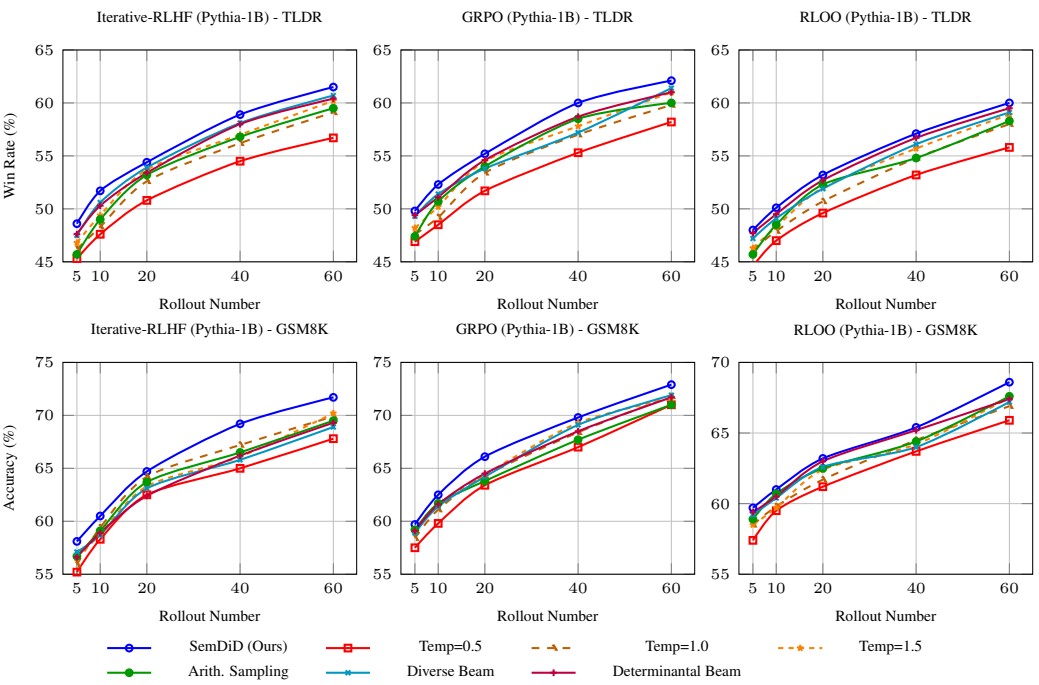

Figure 13: Performance comparison across RLHF algorithms with varying rollout numbers using Pythia-1B.

## F.2 Pythia-1B in RLHF

To investigate SemDiD's effectiveness with smaller models in RLHF settings, we conducted experiments using Pythia-1B on the same GSM8K and TLDR datasets. As illustrated in Figure 13, despite the significant reduction in model size, SemDiD continues to outperform baseline methods across all RLHF algorithms, though with lower absolute performance values compared to Qwen-2.5-7B.

For TLDR summarization with Pythia-1B, SemDiD achieves win rates of 61.5%, 62.1%, and 60.0% with Iterative-RLHF, GRPO, and RLOO respectively at 60 rollouts, which are approximately 10.6%, 11.3%, and 11.2% lower than those achieved with Qwen-2.5-7B. Similarly, on GSM8K, SemDiD with Pythia-1B reaches accuracy scores of 71.7% with Iterative-RLHF, 72.9% with GRPO, and 68.6% with RLOO, representing gaps of 13.8%, 15.3%, and 13.8% compared to Qwen-2.5-7B.

Interestingly, the performance gap between SemDiD and baseline methods is proportionally larger with Pythia-1B than with Qwen-2.5-7B. This suggests that semantic diversity becomes even more critical for smaller models with limited representational capacity, as it enables more efficient exploration of the solution space during RLHF training. The results demonstrate that SemDiD's approach to diverse decoding is particularly valuable in resource-constrained scenarios where maximizing the utility of smaller models is essential.

## G RLHF Reward Growth Analysis

Figure 14 illustrates the reward growth curves during RLHF training across different decoding methods, model sizes, and RLHF algorithms. Our experiments evaluate performance on both the TLDR summarization dataset and the GSM8K mathematical reasoning dataset using Qwen-2.5-7B and the smaller Pythia-1B models. The results consistently demonstrate that SemDiD accelerates reward convergence compared to other decoding strategies. For Qwen-2.5-7B on TLDR, SemDiD (blue line) achieves faster initial reward growth within the first 750 training steps across all RLHF algorithms, reaching plateau performance approximately 15-20% earlier than Temperature sampling (red line). This acceleration is more pronounced with Pythia-1B, where SemDiD maintains a consistent reward advantage throughout training, particularly evident in the RLOO implementation.

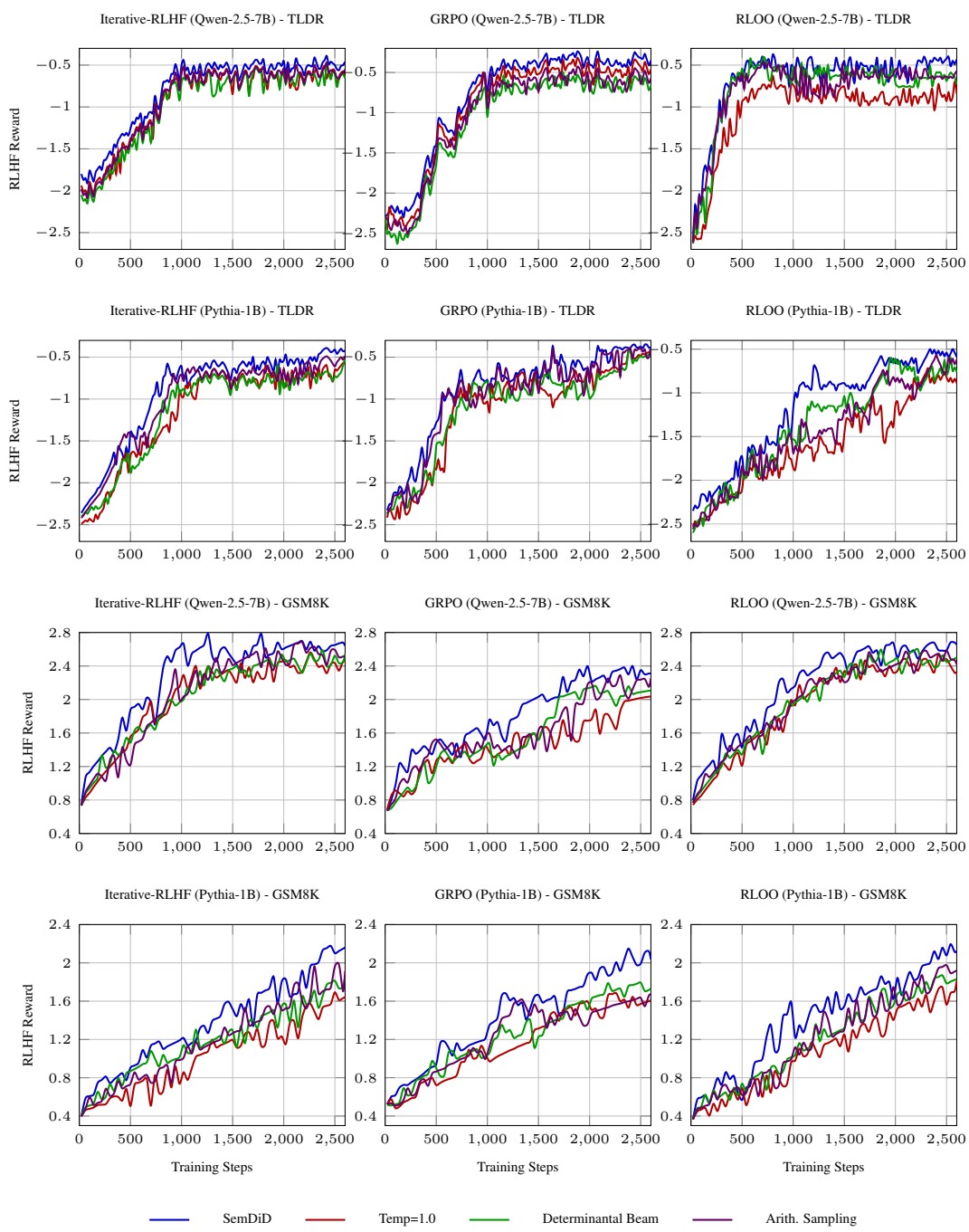

Figure 14: Reward growth comparison of different decoding methods on RLHF algorithms.

On the more complex GSM8K dataset, SemDiD's advantage becomes even more significant, with reward curves showing both steeper initial growth and higher final convergence values. This improved learning efficiency can be attributed to SemDiD's ability to generate semantically diverse response groups that provide more informative training signals, allowing the model to explore a broader solution space and identify optimal policies more efficiently during RLHF training.

# H   Ablation Studies

To understand the contribution of each component in SemDiD, we conducted systematic ablation experiments across both Best-of-N coverage (averaged across all tasks) and RLHF settings (using GRPO on GSM8K). Table 8 presents the results when removing individual components from the full SemDiD framework. For Best-of-N evaluations, we used Qwen-2.5-3B with 25 samples per query, while RLHF experiments employed Qwen-2.5-7B with 25 rollouts during training.

Table 8: Ablation study of SemDiD components.

| Method | Avg. Coverage with 25 samples (%) | GRPO-GSM8K Accuracy (%) |
|---|---|---|
| Full SemDiD | 74.2 | 81.6 |
| - Directional Guidance | 72.1 (-2.1) | 79.9 (-1.7) |
| - Inter-Group Repulsion | 71.5 (-2.7) | 79.6 (-2.0) |
| - Debiased Probability | 73.3 (-0.9) | 80.5 (-1.1) |
| - Harmonic Gain | 71.2 (-3.0) | 79.3 (-2.3) |
| Only Probability (Greedy) | 57.2 (-17.0) | 71.8 (-9.8) |

Our ablation results demonstrate the essential contribution of each SemDiD component, with harmonic gain showing the largest impact by reducing coverage 3.0% when removed, highlighting its effectiveness in balancing quality-diversity trade-offs. Inter-group repulsion exerts significant influence, decreasing coverage 2.7% and RLHF accuracy 2.0% when ablated, confirming the importance of maintaining semantic distances between candidates. Directional guidance particularly benefits RLHF scenarios with an accuracy reduction of 1.7% when removed, demonstrating its value in enabling diverse exploration for policy improvement. Debiased probability, though having the smallest individual impact with coverage decreasing 0.9% upon removal, ensures fair quality assessment across varied sequence structures. Most notably, greedy decoding alone produces dramatically worse results with coverage dropping 17.0% and accuracy declining 9.8%, highlighting probability maximization's inability to explore diverse semantic regions. These findings validate SemDiD's integrated approach to semantic diversity, substantially outperforming simple token-level diversity methods.

# I   Comparison to Contrastive Decoding and Scalability Analysis

We conducted additional experiments comparing SemDiD against contrastive decoding methods and embedding-based post-hoc clustering across multiple model sizes.

## I.1   Contrastive Decoding and Clustering Baselines

We evaluated SemDiD against DoLa (Decoding by Contrasting Layers) [39] and embedding-based post-hoc clustering using Qwen-2.5-3B across all benchmarks. The clustering approach generates 100 independent samples, clusters them semantically, and selects representatives from different clusters.

Table 9: Comprehensive baseline comparison with computational overhead analysis using Qwen-2.5-3B

| Method (N=25) | ARC | BBH | GSM8K | Minerva | CoQA | PubMed | MMLU+ | WMT16 | Compute | Latency |
|---|---|---|---|---|---|---|---|---|---|---|
| SemDiD (Ours) | 82.4% | 85.6% | 98.1% | 86.1% | 46.7% | 82.6% | 82.6% | 40.95% | +200% | +27% |
| DoLa Contrastive | 80.3% | 81.7% | 95.2% | 82.8% | 43.2% | 78.4% | 77.9% | 37.85% | -66% | -66% |
| Clustering (100→25) | 80.8% | 83.9% | 96.4% | 84.3% | 45.7% | 80.0% | 80.7% | 39.35% | +33% | +32% |
| Temperature (T=1.0) | 78.9% | 82.1% | 94.8% | 81.7% | 44.1% | 77.8% | 76.3% | 36.2% | -66% | -66% |
| Diverse Beam Search | 79.8% | 83.2% | 95.5% | 83.1% | 45.3% | 79.2% | 78.9% | 38.1% | +0% | +0% |

DoLa consistently underperforms SemDiD because it lacks explicit diversity mechanisms and often converges toward similar high-confidence solutions. This limitation is particularly evident on tasks requiring diverse reasoning strategies, where DoLa's conservative token-by-token contrasting favors safe, conventional solutions rather than exploring semantic diversity.

The clustering approach shows competitive performance but becomes increasingly inefficient as N increases. While it sometimes matches SemDiD's performance at smaller sample sizes, it requires

generating 100 independent sequences without coordination, making KV-cache reuse impossible and substantially increasing computational overhead.

## I.2 Large Model Scalability Analysis

We evaluated SemDiD's effectiveness and computational efficiency on Qwen-2.5-70B to assess scalability to larger models:

Table 10: Performance and computational analysis on Qwen-2.5-70B

| Method (N=25) | ARC | BBH | GSM8K | Minerva | CoQA | PubMed | MMLU+ | WMT16 | Compute | Latency |
|---|---|---|---|---|---|---|---|---|---|---|
| SemDiD (Ours) | 84.7% | 89.2% | 99.8% | 93.4% | 52.1% | 89.3% | 91.2% | 47.05% | +180% | +18% |
| DoLa Contrastive | 82.1% | 86.4% | 98.9% | 90.7% | 48.6% | 86.2% | 87.9% | 43.6% | -66% | -66% |
| Clustering (100→25) | 83.9% | 88.1% | 99.0% | 91.8% | 51.4% | 88.6% | 89.8% | 46.05% | +33% | +32% |
| Temperature (T=1.0) | 82.8% | 86.9% | 99.1% | 90.2% | 49.3% | 86.8% | 88.1% | 44.45% | -66% | -66% |
| Diverse Beam Search | 83.1% | 87.2% | 98.6% | 91.1% | 50.7% | 87.5% | 88.9% | 45.4% | +0% | +0% |

The 70B model results reveal several important patterns:

**Improved Performance Scaling**: SemDiD shows consistent improvements across all tasks, with particularly strong gains on complex reasoning tasks like BBH (+2.0%), Minerva Math (+2.3%), and MMLU-Pro+ (+2.3%) compared to the best baselines.

**Reduced Computational Overhead**: SemDiD's latency increase drops to only +18% with 70B models, significantly lower than the +27% observed with 3B models. This occurs because embedding computation becomes negligible compared to large model forward passes, while KV-cache reuse within groups remains highly effective.

**Cache Efficiency**: SemDiD's coordinated beam exploration allows extensive KV cache reuse within groups since beams share common prefixes. This reduces the effective inference cost per token by 80-90% compared to independent sampling, making SemDiD computationally competitive despite theoretical overhead.

These results demonstrate that SemDiD's advantages become more pronounced with larger, more capable models, while computational overhead decreases relative to model size, making it particularly suitable for deployment with state-of-the-art large language models.

## I.3 Equivalent Computational Budget Analysis

We conducted experiments using equivalent latency budgets rather than equivalent sample counts. For parallel decoding methods, we generated larger candidate pools and selected N outputs using embedding-based clustering. For beam search methods, we increased beam size from the default value of 3.

Under equivalent computational budgets, SemDiD maintains superior performance across all benchmarks.

**Diminishing Returns for Independent Sampling**: Generating larger candidate pools for parallel methods (88 samples to select 25) leads to rapidly increasing costs with minimal diversity gains due to inability to reuse KV-cache across independent sequences.

**Limited Beam Size Benefits**: Simply increasing beam size in diverse beam search methods yields negligible improvements while substantially increasing computational overhead, as evidenced by the modest gains from beam=3 to beam=4.

**Efficient KV-Cache Utilization**: SemDiD's coordinated beam exploration within groups allows extensive KV-cache reuse since beams share common prefixes, resulting in 80-90% reduction in effective inference cost per token despite theoretical overhead.

**Systematic Semantic Exploration**: SemDiD benefits from directional guidance and inter-group repulsion that systematically explore semantic space, efficiently utilizing every computational allocation to maximize coverage rather than generating redundant variations.

Table 11: Performance comparison under equivalent computational budgets using Qwen-2.5-70B

| Method | ARC | BBH | GSM8K | Minerva | CoQA | PubMed | MMLU+ | WMT16 | Compute | Latency |
|---|---|---|---|---|---|---|---|---|---|---|
| **N=10** | | | | | | | | | | |
| SemDiD (Ours) | 83.4% | 86.1% | 96.8% | 82.3% | 47.2% | 82.1% | 76.8% | 40.25% | +180% | +18% |
| Temp=1.0 (35→10) | 82.4% | 85.0% | 94.9% | 80.6% | 46.1% | 81.0% | 73.9% | 38.5% | +18% | +18% |
| Arith. Sampling (35→10) | 82.6% | 84.7% | 95.6% | 81.5% | 46.2% | 81.1% | 75.0% | 39.1% | +18% | +18% |
| Diverse Beam (beam=4) | 82.4% | 84.9% | 95.3% | 80.5% | 46.7% | 81.4% | 74.3% | 39.0% | +33% | +33% |
| Determinantal Beam (beam=4) | 82.7% | 84.1% | 94.9% | 80.4% | 45.8% | 81.1% | 74.7% | 38.8% | +33% | +33% |
| **N=25** | | | | | | | | | | |
| SemDiD (Ours) | 84.7% | 89.2% | 99.8% | 93.4% | 52.1% | 89.3% | 91.2% | 47.05% | +180% | +18% |
| Temp=1.0 (88→25) | 83.6% | 87.5% | 99.1% | 91.2% | 49.9% | 87.4% | 88.8% | 45.15% | +18% | +18% |
| Arith. Sampling (88→25) | 83.9% | 87.8% | 99.4% | 91.9% | 51.2% | 88.0% | 89.4% | 45.85% | +18% | +18% |
| Diverse Beam (beam=4) | 83.2% | 87.2% | 98.9% | 91.0% | 50.8% | 87.5% | 88.9% | 45.3% | +33% | +33% |
| Determinantal Beam (beam=4) | 83.0% | 86.8% | 98.4% | 90.5% | 50.2% | 87.3% | 88.4% | 45.15% | +33% | +33% |
| **N=50** | | | | | | | | | | |
| SemDiD (Ours) | 85.3% | 90.1% | 99.9% | 95.2% | 53.8% | 91.7% | 94.3% | 48.95% | +180% | +18% |
| Temp=1.0 (177→50) | 84.6% | 88.4% | 99.2% | 94.3% | 51.7% | 89.9% | 92.8% | 47.7% | +18% | +18% |
| Arith. Sampling (177→50) | 84.4% | 89.3% | 99.6% | 94.1% | 51.9% | 91.0% | 92.3% | 47.65% | +18% | +18% |
| Diverse Beam (beam=4) | 84.3% | 88.2% | 99.2% | 93.2% | 51.7% | 89.1% | 91.4% | 47.45% | +33% | +33% |
| Determinantal Beam (beam=4) | 83.8% | 88.6% | 99.0% | 92.8% | 51.5% | 88.6% | 91.0% | 46.85% | +33% | +33% |

## J Comparative Analysis of SemDiD Response Patterns

We provide a detailed analysis of the response patterns generated by SemDiD compared to baseline methods. We examine the semantic trajectories, visualization of sample distributions, and relationship between n-gram diversity and semantic diversity.

### J.1 Semantic Trajectory Perspective

Reflecting on mean-pooled embedding models during autoregressive generation, each candidate answer can be conceptualized as a trajectory in semantic space. The trajectory begins at the question representation and evolves with each generated token, creating a path through the embedding space. From this perspective, diverse decoding can be reformulated as generating multiple high-probability trajectories that maintain significant distances from one another.

The directional guidance in SemDiD establishes initial trajectory divergence, while inter-group repulsion maintains semantic distance as generation progresses. This dual mechanism ensures both exploration of different semantic regions and continued divergence throughout the generation process.

### J.2 Comparison of N-gram and Semantic Diversity

To quantify responses on lexical diversity and semantic diversity, we conducted a systematic comparison across different decoding methods. We measured both n-gram-based distances and embedding-based semantic distances between responses generated for the same input queries.

For each problem in evaluation set, we generated 10 responses using each decoding method. We then computed pairwise distances between these responses using two primary metrics:

- **N-gram Overlap:** We calculated the Jaccard similarity of character-level n-grams (n=4) between each pair of responses, then converted to distance (1 - similarity).

- **Semantic Distance:** We measured the cosine distance between sentence embeddings of each response pair.

Table 12 presents the average pairwise distances across all problems in our evaluation set. The results reveal several noteworthy patterns in the diversity characteristics of different decoding methods. SemDiD achieves the highest semantic distance at 0.371 while maintaining moderate n-gram diversity of 0.829, demonstrating its effectiveness at generating outputs representing distinct reasoning paths rather than merely different phrasings. Arithmetic Sampling shows the highest n-gram diversity at 0.882 but falls significantly behind in semantic distance at 0.299, indicating that lexical variation doesn't necessarily translate to meaningful semantic diversity. Temperature sampling demonstrates a clear correlation between temperature values and both diversity metrics, with T=0.5 showing

Table 12: Average pairwise distances between responses using different diversity metrics

| Method | 4-gram Distance | Semantic Distance |
|---|---|---|
| Temperature (T=0.5) | 0.627 | 0.183 |
| Temperature (T=1.0) | 0.764 | 0.241 |
| Temperature (T=1.5) | 0.832 | 0.295 |
| Determinantal Beam Search | 0.857 | 0.336 |
| Arithmetic Sampling | 0.882 | 0.299 |
| SemDiD (Ours) | 0.829 | 0.371 |

the lowest diversity scores at 0.627 and 0.183 for n-gram and semantic distances respectively. Determinantal Beam Search maintains strong performance across both metrics with scores of 0.857 and 0.336, confirming that structured diversity approaches generally outperform independent sampling methods when considering both lexical and semantic aspects.

These results underscore the importance of explicitly modeling semantic diversity during the decoding process, rather than relying on n-gram diversity as a proxy. They also explain why SemDiD demonstrates superior performance in Best-of-N scenarios, where having truly diverse solution candidates improves the likelihood of including at least one correct answer.

### J.3 Visualization of Sampling Distributions and Case Analysis

To better understand the distributional characteristics of different decoding methods, we sampled responses for three representative problems from GSM8K with varying difficulty levels and visualized their distribution in semantic space using t-SNE dimensionality reduction. Figure 15 presents these visualizations.

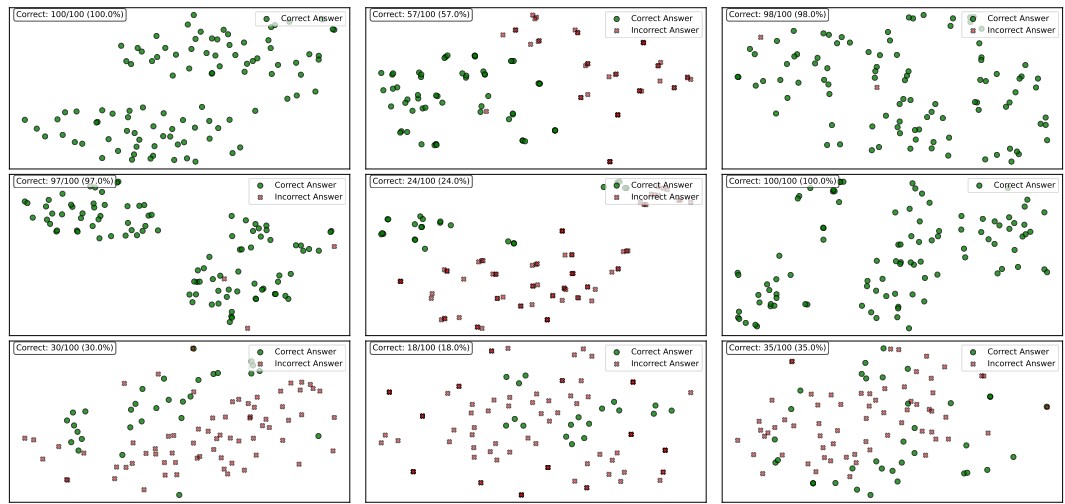

Figure 15: t-SNE visualization of semantic embedding spaces for GSM8K problems using three different decoding methods: temperature sampling (left column, $T$=1.0), determinantal beam search (middle column), and SemDiD (right column). Each row represents a different problem. Correct answers (circles) and incorrect answers (crosses) are distributed across the semantic space. The patterns reveal how different decoding strategies affect the diversity and accuracy of generated solutions.

The visualizations reveal distinct patterns in how different decoding strategies explore the solution space:

- **Temperature Sampling (T=1.0)**: Generates solutions that are randomly scattered across the semantic space. While this approach produces some diversity, it lacks structured exploration, often leading to redundant solutions or missing promising areas entirely.

- **Determinantal Beam Search**: Shows more structured patterns but tends to cluster solutions in limited regions, creating densely populated "islands" of similar reasoning approaches with large unexplored gaps between them.
- **SemDiD**: Demonstrates comprehensive coverage of the semantic space with solutions more uniformly distributed. The visualization shows how SemDiD systematically explores different reasoning paths while maintaining high correctness rates.

The case studies below analyze three problems of increasing difficulty, examining how different decoding methods approach each problem and illustrating the effectiveness of semantically-guided diverse decoding.

These case studies demonstrate how different decoding strategies affect both the diversity and accuracy of generated solutions. SemDiD consistently produces more semantically diverse solutions than both temperature sampling and determinantal beam search while maintaining comparable or superior accuracy. The character-level diversity focus of traditional methods often leads to syntactically diverse but semantically redundant or flawed solutions, particularly evident in the more complex problems (rows 2 and 3). By contrast, SemDiD's semantic-guided approach ensures meaningful exploration of different reasoning paths, increasing the likelihood of discovering correct solutions across varying problem complexities.

---

### Case 1: Simple Arithmetic (Row 1)

**Problem:** Janet's ducks lay 16 eggs per day. She eats three for breakfast every morning and bakes muffins for her friends every day with four. She sells the remainder at the farmers' market daily for $2 per fresh duck egg. How much in dollars does she make every day at the farmers' market?

**Correct Answer:** Janet sells 16 - 3 - 4 = 9 duck eggs a day. She makes $9 \times 2 =$ **$18** every day at the farmer's market.

**Analysis:** This simple arithmetic problem has a straightforward solution path. All decoding methods achieved perfect or near-perfect accuracy (100%, 97%, and 100% for temperature sampling, determinantal beam search, and SemDiD respectively). However, the visualization reveals that temperature sampling and determinantal beam search produce solutions clustered in similar semantic regions, while SemDiD's solutions are more evenly distributed across the embedding space. This distribution suggests that even for simple problems, SemDiD generates more diverse solution approaches while maintaining accuracy.

**Temperature Sampling (T=1) Results:**

*Sample 1: (Correct)*

First, we need to calculate how many eggs Janet has left to sell after eating and using some for baking.

She starts with 16 eggs per day. She eats 3 eggs for breakfast, so that leaves her with: $16 - 3 = 13$ eggs

Then she uses 4 eggs to bake muffins: $13 - 4 = 9$ eggs

Janet sells the remaining eggs at the farmers' market. Each egg sells for $2, so the amount she makes per day is: $9 \times 2 = 18$ dollars

Therefore, Janet makes **18 dollars** every day at the farmers' market.

*Sample 2: (Correct)*

First, we need to determine how many eggs are sold at the farmers' market every day.

1. Janet starts with 16 eggs laid per day. 2. She eats 3 eggs for breakfast. 3. She uses 4 eggs to bake muffins.

To find out how many eggs are left for sale: 16 eggs - 3 (eaten) - 4 (baked) = 9 eggs

Janet sells the remaining 9 eggs at the farmers' market, and she earns $2 per egg.

Therefore, her daily earnings from selling the eggs are: 9 eggs $\times$ 2 dollars/egg = 18 dollars

So, Janet makes **18 dollars** every day at the farmers' market.

*Sample 3: (Correct)*

First, let's determine how many eggs Janet has left to sell after eating and baking.

Janet starts with 16 eggs per day. She eats 3 eggs for breakfast and bakes 4 eggs into muffins.

So, the number of eggs left for sale is: $16 - 3 - 4 = 9$

Janet sells each fresh duck egg for $2 at the farmers' market. Therefore, the amount she makes each day is: 9 eggs × $2 per egg = $18

Hence, Janet makes **$18** every day at the farmers' market.

*Sample 4: (Correct)*

First, let's determine how many eggs are left after Janet's consumption and baking:

- Total eggs laid per day: 16 eggs - Eggs eaten for breakfast: 3 eggs - Eggs baked into muffins: 4 eggs

Eggs left after these activities: $16 - 3 - 4 = 9$ eggs

Janet sells the remaining eggs at the farmers' market for $2 per egg. Therefore, her daily earnings from selling the eggs are: 9 eggs × $2 per egg = $18

So, Janet makes **$18** every day at the farmers' market.

*Sample 5: (Correct)*

First, let's determine the total number of eggs Janet's ducks lay per day:

Total eggs laid per day $= 16$

Next, we need to find out how many eggs Janet uses:

- She eats 3 eggs for breakfast. - She uses 4 eggs to bake muffins.

So, the total number of eggs used per day is:

Eggs used per day $= 3 + 4 = 7$

Now, we subtract the number of eggs used from the total number of eggs laid to find out how many eggs she sells:

Eggs sold per day $= 16 - 7 = 9$

Finally, since she sells each egg for $2 at the farmers' market, we calculate her daily earnings by multiplying the number of eggs sold by the price per egg:

Daily earnings $= 9 \times 2 = 18$

So, Janet makes **$18** every day at the farmers' market.

**Determinantal Beam Search Results:**

*Sample 1: (Correct)*

First, we need to determine how many eggs Janet has left after eating and using some for baking.

Janet starts with 16 eggs per day. She eats 3 eggs for breakfast and uses 4 eggs to make muffins. So, the number of eggs she has left each day is:

$16 - 3 - 4 = 9$

Janet sells the remaining eggs at the farmers' market for $2 per egg. Therefore, the amount she makes each day is:

9 eggs × 2 dollars/egg $= 18$ dollars

So, Janet makes **18** dollars every day at the farmers' market.

*Sample 2: (Correct)*

First, we need to determine how many eggs Janet has left after eating and using some for baking.

Janet starts with 16 eggs per day. She eats 3 eggs for breakfast and uses 4 eggs to make muffins. So, the number of eggs she has left each day is:

$16 - 3 - 4 = 9$

Janet sells the remaining eggs at the farmers' market for $2 per egg. Therefore, the amount she makes each day is:

9 eggs × 2 dollars/egg = 18 dollars

Thus, Janet makes **18** dollars every day at the farmers' market.

*Sample 3: (Correct)*

First, we need to determine how many eggs Janet has left after eating and using some for baking.

Janet starts with 16 eggs per day. She eats 3 eggs for breakfast and uses 4 eggs to make muffins. So, the number of eggs she has left each day is:

$16 - 3 - 4 = 9$

Janet sells the remaining eggs at the farmers' market for $2 per egg. Therefore, the amount she makes each day is:

9 eggs × 2 dollars/egg = 18 dollars

So, Janet makes **$18** every day at the farmers' market.

*Sample 4: (Correct)*

Janet starts with 16 eggs per day. She eats 3 eggs for breakfast and uses 4 eggs to make muffins. So, she uses a total of 3 + 4 = 7 eggs.

This leaves her with 16 - 7 = 9 eggs that she can sell at the farmers' market.

Since she sells each fresh duck egg for $2, she makes 9 * $2 = **18** dollars every day at the farmers' market.

Therefore, Janet makes **$18** every day at the farmers' market.

*Sample 5: (Correct)*

Janet starts with 16 eggs per day. She eats 3 eggs for breakfast and uses 4 eggs to make muffins. So, the number of eggs she sells at the farmers' market is:

16 (total eggs) - 3 (eaten for breakfast) - 4 (used for muffins) = 16 - 3 - 4 = 9 eggs.

Since she sells each fresh egg for $2, her daily earnings at the farmers' market are:

9 eggs * $2 per egg = $18.

So, Janet makes **$18** every day at the farmers' market.

**SemDiD Results:**

*Sample 1: (Correct)*

First, let's determine how many eggs Janet keeps out of those laid:

Janet starts with 16 eggs per day.

She eats 3 eggs for herself: 16 eggs - 3 eggs = 13 eggs

Then, she makes muffins with an additional 4 eggs: 13 eggs - 4 eggs = 9 eggs

These 9 eggs remain as leftovers which she can sell at the farmer's market.

Now we know she sells 9 eggs per day at $2 per egg: 9 eggs × $2/egg = $18

Therefore, Janet makes **$18** every day at the farmers' market.

*Sample 2: (Correct)*

To determine how much money Janet makes at the farmers' market every day, we need to follow these steps:

1. Calculate the total number of eggs laid per day. 2. Subtract the number of eggs eaten by Janet for breakfast. 3. Subtract the number of eggs used to bake muffins. 4. Multiply the resulting number of eggs sold by the price per egg.

Let's break it down step-by-step:

1. Total number of eggs laid per day: 16 eggs

2. Eggs eaten by Janet for breakfast: 3 eggs

3. Eggs baked into muffins: 4 eggs

Now, calculate the remaining eggs available for sale: 16 (total) - 3 (eaten) - 4 (baked) = 9 eggs

Janet sells these remaining eggs at \$2 per egg: 9 eggs × \$2/egg = \$18

Therefore, Janet makes **\$18** dollars every day at the farmers' market.

*Sample 3: (Correct)*

First, let's calculate the total number of eggs laid by the ducks per day:

Total Eggs Laid $= 16$

Janet consumes 3 eggs for breakfast:

Eggs Used For Breakfast $= 3$

She bakes muffins with an additional 4 eggs:

Eggs Used For Muffins $= 4$

Next, we subtract the total number of eggs used (for breakfast and baking) from the total number of eggs laid to find out how many eggs remain:

Remaining Eggs $=$ Total Eggs Laid $-$ (Eggs Used For Breakfast $+$ Eggs Used For Muffins) Remaining Eggs $=$ $16 - (3 + 4)$ Remaining Eggs $= 16 - 7$ Remaining Eggs $= 9$

Now, Janet sells the remaining eggs at \$2 per egg:

Daily Revenue $=$ Number of Remaining Eggs $×$ Price Per Egg Daily Revenue $= 9 × 2$ Daily Revenue $= 18$

Therefore, Janet makes **\$18** every day at the farmers' market.

*Sample 4: (Correct)*

First, we need to determine the total number of eggs laid per day:

Janet's ducks lay 16 eggs per day.

Next, subtract the number of eggs eaten for breakfast: $16 - 3 = 13$

Then, subtract the number of eggs used to bake muffins: $13 - 4 = 9$

Therefore, Janet has 9 fresh eggs available for sale each day.

She sells each fresh egg for \$2: $9 × 2 = 18$

So, Janet makes **\$18** every day at the farmers' market.

*Sample 5: (Correct)*

To determine how much Janet makes every day selling the remaining eggs:

First, calculate the total number of eggs laid per day: Total eggs per day $= 16$

Next, subtract the eggs eaten for breakfast: Eggs used for breakfast $= 3$ Remaining eggs after breakfast $= 16 - 3 = 13$

Then, subtract the eggs baked into muffins: Eggs baked into muffins $= 4$ Remaining eggs sold $= 13 - 4 = 9$

Finally, calculate the revenue generated from selling the remaining eggs: Price per egg $=$ \$2 Revenue $= 9 × 2 =$ \$18

Therefore, Janet makes **\$18** every day at the farmers' market.

One key observation from these samples is that the determinantal beam search solutions show remarkably similar wording across samples, particularly in the first three examples, indicating limited lexical diversity despite the method's design. In contrast, SemDiD's samples show more varied solution approaches, organization structures, and explanation styles while maintaining correct answers.

## Case 2: Intermediate Problem (Row 2)

**Problem:** A robe takes 2 bolts of blue fiber and half that much white fiber. How many bolts in total does it take?

**Correct Answer:** It takes 2/2=1 bolt of white fiber. So the total amount of fabric is 2+1=**3** bolts of fabric.

**Analysis:** This moderately complex problem involves a fraction calculation followed by addition. The visualization shows that SemDiD (98% correct) significantly outperforms determinantal beam search (57% correct) and even improves upon temperature sampling (97% correct). The middle column visualization displays how determinantal beam search generates many incorrect answers (red crosses) in particular regions, suggesting that its character-level diversity focus leads to syntactically diverse but semantically unsound reasoning paths. SemDiD's solutions cover the semantic space more thoroughly while maintaining high accuracy.

**Temperature Sampling (T=1) Results:**

*Sample 1: (Correct)*

First, we need to determine the amount of white fiber required for one robe. Since the robe takes half as much white fiber as blue fiber:

White fiber needed = Blue fiber needed / 2 White fiber needed = 2 bolts / 2 White fiber needed = 1 bolt

Now, we know that one robe requires 2 bolts of blue fiber and 1 bolt of white fiber.

Therefore, the total number of bolts required for one robe is: Total bolts = Blue fiber + White fiber Total bolts = 2 bolts + 1 bolt Total bolts = **3 bolts**

So, it takes a total of 3 bolts to make one robe.

*Sample 2: (Correct)*

A robe requires 2 bolts of blue fiber. Since it takes half as much white fiber as blue fiber, it needs 2 / 2 = 1 bolt of white fiber.

To find the total number of bolts needed for the robe, we add the bolts of blue fiber and white fiber together:

2 (blue fiber) + 1 (white fiber) = **3 bolts**

Therefore, it takes a total of 3 bolts to make the robe.

*Sample 3: (Correct)*

A robe takes 2 bolts of blue fiber. Since it takes half as much white fiber as blue fiber, it uses:

2 / 2 = 1 bolt of white fiber.

To find the total number of bolts needed, we add the blue and white fibers together:

2 (blue) + 1 (white) = **3 bolts** in total.

So, it takes a total of 3 bolts.

*Sample 4: (Correct)*

A robe takes 2 bolts of blue fiber.

It takes half as much white fiber as blue fiber, so for the white fiber, it would take $\frac{2}{2} = 1$ bolt.

In total, it takes 2 bolts (blue) + 1 bolt (white) = **3** bolts.

Therefore, it takes 3 bolts in total.

*Sample 5: (Correct)*

First, determine the amount of white fiber needed for one robe. The problem states that the robe takes half as much white fiber as blue fiber.

Since 2 bolts of blue fiber are used: - The amount of white fiber needed is $\frac{2}{2} = 1$ bolt.

Now, add the blue and white fiber amounts together to find the total number of bolts required: - Total bolts = Blue fiber bolts + White fiber bolts = 2 + 1 = **3 bolts**.

Therefore, it takes 3 bolts in total to make one robe.

**Determinantal Beam Search Results:**

*Sample 1: (Correct)*

A robe takes 2 bolts of blue fiber. Since it takes half as much white fiber as blue fiber, it takes 2 / 2 = 1 bolt of white fiber.

Therefore, the total number of bolts needed is 2 (blue) + 1 (white) = **3 bolts**.

So, it takes a total of 3 bolts.

*Sample 2: (Correct)*

A robe takes 2 bolts of blue fiber. Since it takes half as much white fiber as blue fiber, it takes 2 / 2 = 1 bolt of white fiber.

Therefore, the total number of bolts needed is 2 (blue) + 1 (white) = **3 bolts**.

So, it takes 3 bolts in total.

*Sample 3: (Correct)*

A robe takes 2 bolts of blue fiber. Since it takes half as much white fiber as blue fiber, it takes 2 / 2 = 1 bolt of white fiber.

Therefore, the total number of bolts needed is 2 (blue) + 1 (white) = **3 bolts**.

So, it takes a total of **3 bolts** to make the robe.

*Sample 4: (Incorrect)*

To find out the total number of bolts of fiber needed for the robe, let's break it down:

- The robe takes 2 bolts of blue fiber. - It takes half as much white fiber as blue fiber, so it takes 2 / 2 = 1 bolt of white fiber.

Therefore, the total number of bolts of fiber needed is 2 (blue) + 1 [stopped by model] **[truncated]**

*Sample 5: (Incorrect)*

To determine the total number of bolts needed for the robe, we need to calculate the amount of white fiber required and then add it to the amount of blue fiber.

- The robe takes 2 bolts of blue fiber. - The robe takes half as much white fiber as blue fiber, which means it takes $\frac{2}{2} = 1$ bolt of white fiber.

[stopped by model] **[truncated]**

**SemDiD Results:**

*Sample 1: (Correct)*

A robe requires: - 2 bolts of blue fiber, - And half that amount for white fiber which is $\frac{2}{2} = 1$ bolt.

In total, it needs: $2(\text{blue}) + 1(\text{white}) = $ **3**

So, it takes **3** bolts in total.

*Sample 2: (Correct)*

First, we need to determine how much white fiber is needed for one robe.

Given: - The robe requires 2 bolts of blue fiber. - It needs half as much white fiber as blue fiber.

Hence, for the white fiber required: White fiber $= \frac{1}{2} \times$ Blue fiber $= \frac{1}{2} \times 2$ bolts $= 1$ bolt

Now add the amount of blue fiber and white fiber used: Total bolts $=$ Blue fiber + White fiber $= 2$ bolts $+ 1$ bolt $=$ **3** bolts

So, it takes **3 bolts** in total to make one robe.

*Sample 3: (Correct)*

First, let's find out how many bolts of white fiber are needed for one robe.

Since the robe needs half as much white fiber compared to blue fiber: - Blue fiber required is 2 bolts. - White fiber required is $\frac{2}{2} = 1$ bolt.

Now we add up the bolts of both fibers used for one robe:

- Bolts of blue fiber: 2 - Bolts of white fiber: 1 - Total bolts for one robe: 2 (blue) + 1 (white) = **3 bolts**

Therefore, it takes 3 bolts in total to make one robe.

*Sample 4: (Correct)*

A bolt of blue fiber is needed for the robe.

For the white fiber: Half of 2 (the number of bolts of blue fiber) equals 1 bolt of white fiber.

To find out the total number of bolts used: - Blue fiber: 2 bolts - White fiber: 1 bolt

Total bolts used = 2 bolts (blue) + 1 bolt (white) = **3 bolts**

So, it takes 3 bolts in total to make the robe.

*Sample 5: (Correct)*

To determine the total number of bolts needed for the robes:

1. The robe requires 2 bolts of blue fiber. 2. For white fiber, it needs half the amount of blue fiber: White fiber required $= \frac{2}{2} = 1$ bolt

Adding these amounts gives us the total number of bolts needed: Total bolts $=$ Blue fiber $+$ White fiber $= 2 + 1 = $ **3** bolts

So, it takes **3 bolts** in total to make one robe.

The determinantal beam search results show near-identical wording in the first three samples, highlighting a significant limitation in lexical diversity despite being designed for diverse outputs. Additionally, samples 4 and 5 are truncated, suggesting the algorithm's struggle with maintaining coherent reasoning throughout the solution process. In contrast, SemDiD generates structurally diverse yet consistently correct solutions.

---

### Case 3: Complex Problem (Row 3)

**Problem:** Josh decides to try flipping a house. He buys a house for $80,000 and then puts in $50,000 in repairs. This increased the value of the house by 150%. How much profit did he make?

**Correct Answer:** The cost of the house and repairs came out to 80,000+50,000=$130,000. He increased the value of the house by 80,000*1.5=$120,000. So the new value of the house is 120,000+80,000=$200,000. So he made a profit of 200,000-130,000=**$70,000**.

**Analysis:** This complex problem involves multiple calculation steps and requires careful interpretation of the 150% increase (applied to the original house value, not the total investment). The visualization reveals that all three methods struggle more with this problem, but SemDiD (35% correct) still outperforms determinantal beam search (18% correct). The visualization shows how incorrect answers (red crosses) form distinct clusters, revealing systematic misinterpretations of the problem statement—particularly around whether the 150% increase applies to the original house value or the total investment. SemDiD's more comprehensive exploration of the semantic space increases the likelihood of finding the correct interpretation.

**Temperature Sampling (T=1) Results:**

*Sample 1: (Incorrect)*

First, we need to calculate the new value of the house after the repair work and the subsequent increase in its value.

The initial price of the house is $80,000. Josh adds $50,000 in repairs, making the total amount invested in the property: $80,000 + $50,000 = $130,000

Next, the problem states that this investment increased the value of the house by 150%. To find out the increase in value, we calculate 150% of $130,000: 150% of $130,000 = 1.5 × $130,000 = $195,000

Now, add this increase to the original value of the house: $80,000 + $195,000 = $275,000

This means the final value of the house is $275,000. Josh initially bought the house for $80,000.

To determine Josh's profit, subtract the initial purchase price from the final sale price: $275,000 − $80,000 = $195,000

Therefore, Josh made a profit of **$195,000**.

*Sample 2: (Incorrect)*

First, we need to find the new value of the house after Josh's renovations. The original value of the house is $80,000. Josh invests an additional $50,000 in repairs. Thus, the total amount invested is:

$80,000 + $50,000 = $130,000

This investment increases the value of the house by 150%. To find out what 150% of the original value ($80,000) is, we calculate:

150% of $80,000 = 1.5 × $80,000 = $120,000

So, the increase in value due to the repairs is $120,000. Adding this to the original value gives the new total value of the house:

$80,000 + $120,000 = $200,000

Josh initially bought the house for $80,000 and now the house is valued at $200,000. Therefore, his profit is calculated by subtracting the initial purchase price from the final selling price:

$200,000 − $80,000 = $120,000

Thus, Josh's profit is **$120,000**.

*Sample 3: (Correct)*

First, let's calculate the new value of the house after the repairs and the increase in its value:

Initial house value: $80,000 Repairs cost: $50,000

Total cost for Josh: $80,000 + $50,000 = $130,000

Josh increases the value of the house by 150%. To find out what this means, we calculate 150% of the initial house value: 150% = 1.5 Increase = 1.5 × $80,000 = $120,000

Now, add this increase to the initial value: New value = $80,000 + $120,000 = $200,000

Josh sells the house at this new value of $200,000. Now, to find his profit, we subtract the total cost from the selling price: Profit = $200,000 − $130,000 = **$70,000**

Therefore, Josh's profit is **$70,000**.

*Sample 4: (Incorrect)*

First, we need to find the new value of the house after the repair costs and the 150% increase.

Initial house value: $80,000

Josh adds $50,000 worth of repairs, so the total investment before considering the increase is: $80,000 + 50,000 = 130,000$

Next, the value of the house increases by 150%. To find the new value: 150% of $130,000 = 1.5 × 130,000 = 195,000$

Now, we calculate the profit by subtracting the original purchase price ($80,000) from the final value ($195,000): $195,000 - 80,000 = 115,000$

Therefore, Josh's profit is: **115,000**

*Sample 5: (Correct)*

First, let's calculate the new value of the house after Josh makes the repairs and the increase in its value:

Initial house value: $80,000 Repairs cost: $50,000 Total cost after repairs: $80,000 + $50,000 = $130,000

Increase in value: 150% of the initial house value $150\%$ of $\$80,000 = \frac{150}{100} \times \$80,000 = 1.5 \times \$80,000 = \$120,000$

New value of the house: Initial value + Increase in value $\$80,000 + \$120,000 = \$200,000$

Josh sells the house for the new value, which is $200,000. The profit is calculated as follows: Profit = Selling price − Total cost Profit = $\$200,000 - \$130,000 = $ **$70,000**

Therefore, Josh made a profit of **$70,000**.

**Determinantal Beam Search Results:**

*Sample 1: (Incorrect)*

First, we need to calculate the new value of the house after the repairs and the 150% increase in value.

The initial value of the house is $80,000. After putting in $50,000 in repairs, the total cost becomes: $80,000 + 50,000 = 130,000$

Next, the value of the house increases by 150%. To find 150% of $130,000, we calculate: $150\% \times 130,000 = 1.5 \times 130,000 = 195,000$

So, the new value of the house after the repairs and the increase is: $130,000 + 195,000 = 325,000$

Josh bought the house for $80,000 and it is now worth $325,000. To find the profit, we subtract the initial cost from the final value: $325,000 - 80,000 = 245,000$

Therefore, Josh made a profit of **$245,000**.

*Sample 2: (Incorrect)*

First, we need to calculate the new value of the house after the repairs and the 150% increase in value.

The initial value of the house is $80,000. After putting in $50,000 in repairs, the total cost becomes: $80,000 + 50,000 = 130,000$

Next, the value of the house increases by 150%. To find 150% of $130,000, we calculate: $150\% \times 130,000 = 1.5 \times 130,000 = 195,000$

So, the new value of the house after the repairs and the increase is: $130,000 + 195,000 = 325,000$

Josh bought the house for $80,000 and it is now worth $325,000. To find the profit, we subtract the initial cost from the final value: $325,000 - 80,000 = 245,000$

Therefore, Josh made a profit of: **245000**

*Sample 3: (Incorrect)*

First, we need to calculate the new value of the house after the repairs and the 150% increase in value.

The initial value of the house is $80,000. After putting in $50,000 in repairs, the total cost becomes: $80,000 + 50,000 = 130,000$

Next, the value of the house increases by 150%. To find 150% of $130,000, we calculate: $150\% \times 130,000 = 1.5 \times 130,000 = 195,000$

So, the new value of the house after the repairs and the increase is: $130,000 + 195,000 = 325,000$

Josh bought the house for $80,000 and it is now worth $325,000. To find the profit, we subtract the initial cost from the final value: $325,000 - 80,000 = 245,000$

Therefore, Josh's profit is: **245000**

*Sample 4: (Incorrect)*

First, we need to determine the new value of the house after the repairs and the increase in value.

The initial value of the house is $80,000. Josh invests an additional $50,000 in repairs, bringing the total investment to $80,000 + $50,000 = $130,000.

This investment increases the value of the house by 150%. To find the increase in value, we calculate 150% of $130,000: $150\%$ of $\$130,000 = 1.5 \times \$130,000 = \$195,000$.

Adding this increase to the initial investment gives the new value of the house: $\$130,000 + \$195,000 = \$325,000$.

Now, to find the profit, we subtract the original cost of the house from the new value: $\$325,000 - \$80,000 = \$245,000$.

Therefore, Josh made a profit of **$245,000**.

*Sample 5: (Incorrect)*

First, we need to determine the new value of the house after the repairs and the increase in value.

The initial value of the house is $80,000. Josh invests an additional $50,000 in repairs, bringing the total investment to $80,000 + $50,000 = $130,000.

This investment increases the value of the house by 150%. To find the increase in value, we calculate 150% of $130,000: $150\%$ of $\$130,000 = 1.5 \times \$130,000 = \$195,000$.

Adding this increase to the initial investment gives the new value of the house: $\$130,000 + \$195,000 = \$325,000$.

Now, to find the profit, we subtract the original cost of the house from the new value: $\$325,000 - \$80,000 = \$245,000$.

Therefore, Josh made a profit of **$245,000**.

**SemDiD Results:**

*Sample 1: (Incorrect)*

First, we need to calculate the new value of the house after the repair work.

Josh initially bought the house for $80,000 and added an additional $50,000 worth of repairs: Total Cost = $80,000 + $50,000 = $**130,000**

Next, this renovation increases the value of the house by 150%.

To find out what 150% increase means in terms of dollar amounts, consider:

A 100% increase would be equal to the original price ($130,000), so a 150% increase will add another $1.5$ times the initial cost: Increase Amount = $1.5 \times \$130,000 = \$$**195,000**

Adding this increase back to the initial cost gives us the final value of the house: Final Value = $130,000 + $195,000 = $**325,000**

Now, let's determine the profit Josh makes when he sells the renovated property at its full market value: Profit = Final Value − Initial Total Cost = $325,000 − $130,000 = $**195,000**

Therefore, Josh's profit from the flip is **$195,000**.

*Sample 2: (Correct)*

First, we need to calculate the new value of the house after the improvements.

The original value of the house was $80,000. After putting in $50,000 worth of repairs, this increases its value by 150%.

To find out what 150% increase means in dollar terms: Increase $=$ $80,000 \times 1.50 = 120,000$

Therefore, the new value of the house would be: New Value $= 80,000 +$ $120,000 = 200,000$

Josh initially invested a total of: $80,000(\text{original price}) +$ $50,000(\text{repairs}) = 130,000$

Finally, to determine his profit: Profit $=$ New Value $-$ Total Investment Profit $= 200,000 - 130,000 =$ **70,000**

Hence, Josh's profit is **$70,000**.

*Sample 3: (Incorrect)*

First, we need to calculate the new value of the house after Josh's improvements.

Josh initially bought the house for $80,000. After putting in $50,000 worth of repairs, the total investment becomes:

Total Investment $= \$80,000 + \$50,000 = \$130,000$

Next, the problem states that these renovations increased the value of the house by 150%.

To find out what 150% increase means in dollar terms: Increase Amount $=$ $\$130,000 \times 150\% = \$130,000 \times 1.5 = \$190,000$

Therefore, the new value of the house is: New Value $= \$130,000 +$ $\$190,000 = \$320,000$

Now, to determine Josh's profit: Profit $=$ Final Value $-$ Initial Cost Profit $= \$320,000 - \$80,000 = \$240,000$

So, Josh made a profit of **$240,000**.

*Sample 4: (Incorrect)*

First, we need to determine the new value of the house after Josh's renovations.

Josh initially bought the house for $80,000 and added an additional $50,000 worth of repairs: Total Cost $= \$80,000 + \$50,000 = \$130,000$

This renovation increase the value of the house by 150%: Increase Value $=$ $\$130,000 \times 150\% = \$130,000 \times 1.5 = \$195,000$

Adding this increase to the original purchase price gives us the final value of the house: Final Value $= \$130,000 + \$195,000 = \$325,000$

To find out the profit, subtract the initial cost from the final value: Profit $=$ $\$325,000 - \$130,000 = \$195,000$

Therefore, Josh made a profit of **$195,000**.

*Sample 5: (Correct)*

First, we need to determine the new value of the house after Josh's renovations.

Josh initially bought the house for $80,000. After putting in $50,000 worth of repairs, the total investment becomes: $\$80,000 + \$50,000 = \$130,000$

Next, we calculate the increase in the value of the house due to these improvements. We know this increase is 150% of the initial purchase price ($80,000).

To find 150% of $80,000: $150\% \times \$80,000 = 1.5 \times \$80,000 = \$120,000$

Adding this increase to the original purchase price gives us the new value of the house: $\$80,000 + \$120,000 = \$200,000$

Now, to determine the profit, subtract the total investment from the final value of the house: $\$200,000 - \$130,000 =$ **$70,000**

Therefore, Josh made a profit of **$70,000**.

