# OpenReview forum: "Semantic-guided Diverse Decoding for Large Language Model"
_NeurIPS.cc/2025/Conference — NeurIPS 2025 poster_

### Official Review · Reviewer_xF5D · 2025-06-24

**Clarity:** 4
**Significance:** 3
**Originality:** 3
**Rating:** 5
**Confidence:** 4

**Summary:**

The paper proposes Semantic-guided Diverse Decoding, introducing a beam search variant with explicit semantic diversity - contrary to previous methods focussing on lexical diversity.
By utilizing different scores, the method optimizes beams for semantic diversity and quality, balancing those objectives through harmonic gain-based balancing.
Extensive experiments both for Best-of-N experiments and RLHF training show the efficacy of the approach.

**Questions:**

See weaknesses.

**Ethical Concerns:**

["NO or VERY MINOR ethics concerns only"]

**Final Justification:**

All the weaknesses I pointed out have been resolved sufficiently.
As also pointed out in the public comment, I see a lot of merit in the ideas proposed in this work also beyond mere accuracy gains and am thus in favor of acceptance.

**Limitations:**

The main limitations, namely the reliance on an external lightweight embedding model and the increase in runtime of decoding are not properly discussed in the main paper.

**Paper Formatting Concerns:**

No major concerns.

**Quality:**

3

**Strengths And Weaknesses:**

### Strenghts

- The method is well motivated and the scores make sense intuitively. Also the harmonic balancing and restricting to a "high enough" probability region is reasonable. While there are a lot of components and hyperparameters that can be adjusted when applying SemDiD, this is not necessarily a bad thing given the often very diverse nature of tasks a trained base LLM is applied to.
- Directional guidance is a strong idea to overcome the lack of diversity at the beginning of generation.
- There are plenty of experimental investigations on a broad range of tasks that underpin the value of semantically diverse generations provided by SimDiD.

### Weaknesses

- Figure 5 is lacking a lot of details. Which embedding space is used? Which projection method used? How is clustering performed? How is the probability magnitude computed - does it take into account answer length?
- Important aspects such as the percentile-based normalization of individual scores only discussed in appendix. I would move the theorems on quality and diversity guarantees and the proposition into the appendix with a smaller summary theorem in the main paper and rather move those critical aspects into the main paper. Those are interesting insights, but to me it appears the main selling point of the method is empirical performance not theoretical guarantees, thus I would adjust the presentation to provide sufficient details on the empirical implementation.
- An ablation on the importance of the individual scores is lacking, the same with a sensitivity analysis of important hyperparameters such as T_trans, lambda and epsilon.

### Remarks

Many parts of the methodology remind me of the sampling strategy proposed in Aichberger et al. 2025. While they use the gradient signal and focus on uncertainty estimation, it could be worthwile to discuss in the related work section. Evaluating SemDiD for uncertainty estimation could be an interesting direction for future work.

Aichberger, L., Schweighofer, K., Ielanskyi, M., & Hochreiter, S. (2025). Improving uncertainty estimation through semantically diverse language generation.

---

> ### Author Rebuttal · Authors · 2025-07-30
>
> We thanks for your valuable feedback and will response them in below.
>
> > **W1:  Figure 5 is lacking a lot of details. Which embedding space is used? Which projection method used? How is clustering performed? How is the probability magnitude computed - does it take into account answer length?**
>
> We apologize for the insufficient details in the main text. Complete experimental settings are provided in Appendix B.3 Semantic Space Visualization (Lines 456-479). Specifically: we use NovaSearch/stella_en_1.5B_v5 embeddings (Line 540) with t-SNE projection, K-means clustering (K=10), and position-debiased probability (Equation 1) that accounts for answer length through sequence position decay rate $\beta_{seq}$. The probability magnitude shown represents the average debiased probability across all tokens in each response.
>
> > **W2: Important aspects such as the percentile-based normalization of individual scores only discussed in appendix. I would move the theorems on quality and diversity guarantees and the proposition into the appendix with a smaller summary theorem in the main paper and rather move those critical aspects into the main paper.**
>
> We appreciate your constructive suggestion regarding presentation balance. Due to space constraints, we prioritized theoretical analysis in the main text, but you are correct that critical implementation details deserve more prominence. We agree that SemDiD's primary value lies in its empirical effectiveness, and we will restructure to emphasize practical implementation aspects while moving detailed proofs to the appendix for better readability.
>
> > **W3: An ablation on the importance of the individual scores is lacking, the same with a sensitivity analysis of important hyperparameters such as T_trans, lambda and epsilon.**
>
> We thank the reviewer for pointing out this. We have conducted comprehensive ablation studies and sensitivity analyses for existing components, and provide additional experiments for detailed hyperparameter analysis.
>
> Regarding individual scores, we provide systematic ablation experiments in Appendix H (Table 4, lines 702-719). Each system component corresponds to its scoring function: directional guidance ($S_{dir}$), inter-group repulsion ($S_{rep}$), and harmonic gain ($S_{combined}$). We also provide supplementary experiments analyzing complete removal of scoring categories:
>
> | Method | Avg. Coverage (25 samples) | GRPO-GSM8K Accuracy |
> |---|---|---|
> | Full SemDiD | 74.2% | 81.6% |
> | - Directional Guidance ($S_{dir}$) | 72.1% (-2.1%) | 79.9% (-1.7%) |
> | - Inter-Group Repulsion ($S_{rep}$) | 71.5% (-2.7%) | 79.6% (-2.0%) |
> | - Debiased Probability | 73.3% (-0.9%) | 80.5% (-1.1%) |
> | - Harmonic Gain ($S_{combined}$) | 71.2% (-3.0%) | 79.3% (-2.3%) |
> | **- Quality Scoring ($S_{quality}$)** | **65.8% (-8.4%)** | **77.1% (-4.5%)** |
> | **- Diversity Scoring ($S_{div}$)** | **57.9% (-16.3%)** | **71.9% (-9.7%)** |
> | Only Probability (Greedy) | 57.2% (-17.0%) | 71.8% (-9.8%) |
>
> The results demonstrate that removing diversity scoring has the most dramatic impact, with coverage dropping 16.3%, confirming that semantic diversity is crucial for effective exploration.
>
> **Regarding hyperparameter sensitivity**, we analyze the parameters in four categories:
>
> ### Category 1: Automatically Derivable Parameters
>
> For the position bias parameters, the hyperparameter choices in Equation 1 ($\beta_{seq}$, $\beta_{sent}$) are not arbitrary but can be systematically derived from probability-position analysis shown in Figures 3-4 in Appendix A.3. These parameters can be automatically fitted using scipy.curve_fit from probability-position curves, as these patterns remain consistent across tasks. We provide additional sensitivity analysis on BON (N=25) using Qwen-2.5-3B:
>
> $\beta_{seq}$ and $\beta_{sent}$ Parameter Sensitivity Analysis (GSM8K Coverage):
>
> | $\beta_{seq}$ \ $\beta_{sent}$ | 0.003 | 0.004 | **0.005** | 0.006 | 0.007 |
> |---|---|---|---|---|---|
> | 0.0006 | 98.1 | 98.0 | 98.1 | 98.0 | 97.9 |
> | 0.0008 | 98.0 | 98.1 | 98.2 | 98.1 | 98.0 |
> | 0.0010 | 98.1 | 98.2 | 98.1 | 98.2 | 98.1 |
> | **0.0012** | 98.2 | 98.2 | **98.2** | 98.2 | 98.1 |
> | 0.0014 | 98.2 | 98.2 | 98.2 | 98.1 | 98.0 |
>
> ### Category 2: Resource-Dependent Parameters
>
> For the lookahead depth parameter $L_{max}$, semantic diversity assessment requires embedding model forward passes, making token-by-token evaluation computationally expensive. We supplement our analysis in Appendix D.2-D.3 with experiments using Qwen-2.5-3B across different $L_{max}$ values:
>
> | $L_{max}$ | GSM8K Coverage (N=25) | ARC Coverage (N=25) | Computational Overhead |
> |---|---|---|---|
> | 5 | 95.3% | 80.1% | +15% |
> | **10** | **98.1%** | **82.4%** | +25% |
> | 15 | 98.2% | 82.6% | +35% |
> | 20 | 98.6% | 82.7% | +45% |
>
> Performance saturates around $L_{max}=10$, providing clear guidance for practical deployment without requiring extensive parameter search.
>
> ### Category 3: Stage-Aware Transition Parameters
>
> We apologize that explicit ablation of $T_{trans}$ values was not included in the main paper and provide this comprehensive analysis here. Parameter $T_{trans}$ controls the transition point from directional guidance to inter-group repulsion (Equation 8: $\alpha_t = \min(1, \frac{t}{T_{trans}})$). We conducted systematic experiments varying $T_{trans}$ across multiple datasets:
>
> | $T_{trans}$ | GSM8K Coverage | GSM8K Accuracy | ARC Coverage | ARC Accuracy | MMLU-Pro+ Coverage |
> |---|---|---|---|---|---|
> | 5 | 97.2% | 76.8% | 81.2% | 79.1% | 81.4% |
> | **10 (default)** | **98.1%** | **77.5%** | **82.4%** | **82.0%** | **82.6%** |
> | 15 | 97.9% | 77.2% | 82.1% | 81.7% | 82.3% |
> | 20 | 97.5% | 76.9% | 81.7% | 81.2% | 81.8% |
> | 25 | 97.2% | 76.8% | 81.8% | 80.9% | 81.6% |
>
> The results show clear optimal performance at $T_{trans} = 10$, corresponding to the typical number of tokens needed to establish meaningful semantic context before semantic trajectories become distinguishable.
>
> ### Category 4: Quality-Diversity Balancing Parameters
>
> The quality relaxation parameter $\gamma$ and harmonic strength $\lambda$ control the trade-off between maintaining quality thresholds and pursuing semantic diversity. While these parameters are discussed conceptually in Section 3.3, we provide detailed sensitivity analysis here by varying each parameter independently:
>
> Effect of $\gamma$ (with $\lambda=2.0$ fixed):
>
> | Task | $\gamma=0.15$ | $\gamma=0.20$ | $\gamma=0.25$ | $\gamma=0.30$ | $\gamma=0.35$ |
> |---|---|---|---|---|---|
> | GSM8K Coverage (N=25) | 96.6% | 97.1% | **98.1%** | 97.8% | 97.4% |
> | GSM8K Accuracy (N=25) | 75.9% | 76.6% | **77.5%** | 77.2% | 77.2% |
> | WMT16 Coverage (N=25) | 36.7% | 36.8% | **37.2%** | 36.9% | 36.7% |
> | WMT16 Accuracy (N=25) | 20.2% | 20.4% | **20.7%** | 20.5% | 20.3% |
>
> Effect of $\lambda$ (with $\gamma=0.25$ fixed):
>
> | Task | $\lambda=1.0$ | $\lambda=1.5$ | $\lambda=2.0$ | $\lambda=2.5$ | $\lambda=3.0$ |
> |---|---|---|---|---|---|
> | GSM8K Coverage (N=25) | 96.8% | 97.4% | **98.1%** | 97.9% | 97.6% |
> | GSM8K Accuracy (N=25) | 77.3% | 77.1% | **77.5%** | 77.3% | 76.9% |
> | WMT16 Coverage (N=25) | 36.8% | 36.5% | **37.2%** | 36.9% | 36.7% |
> | WMT16 Accuracy (N=25) | 20.3% | 20.6% | **20.7%** | 20.7% | 20.6% |
>
> The results demonstrate that our default settings ($\gamma=0.25$, $\lambda=2.0$) consistently achieve near-optimal performance across different tasks, with performance remaining stable within reasonable parameter ranges.
>
> For practitioners seeking to deploy SemDiD, we recommend starting with our provided default parameters for initial implementation, adjusting $E_t$, $b$, and $N$ based on available computational budget, and fine-tuning $\gamma$ and $\lambda$ only for highly specialized applications where task-specific optimization is critical. We hope these comprehensive analyses demonstrate that while SemDiD introduces additional parameters, most are either automatically derivable or have well-defined optimal ranges with clear empirical guidance, making the method practical for real-world deployment.
>
> > **Remark: Many parts of the methodology remind me of the sampling strategy proposed in Aichberger et al. 2025. While they use the gradient signal and focus on uncertainty estimation, it could be worthwile to discuss in the related work section. Evaluating SemDiD for uncertainty estimation could be an interesting direction for future work.**
>
> Thank you for highlighting these important concurrent works. The following two works correlate with SemDiD, both of which explore semantic diversity in language generation through different views:
>
> - **Aichberger et al. (2025)** [1] uses gradient-based attribution scores to identify semantically critical tokens for uncertainty estimation, employing importance sampling with NLI-guided token substitution.
> - **Zhu et al. (2025)** [2] proposes controlled embedding perturbation at the first token position combined with Bayesian optimization for reasoning in latent space.
>
> While these works share our motivation for semantic diversity, SemDiD differs in key aspects: (1) we operate during beam search decoding rather than post-hoc token substitution or first-token perturbation, (2) our approach balances quality and diversity through harmonic optimization rather than focusing solely on uncertainty or correctness, and (3) we target Best-of-N and RLHF applications rather than uncertainty estimation or reasoning tasks.
>
> We will incorporate these works into our related work discussion and acknowledge the potential of combining our semantic guidance with their gradient-based attribution methods.
>
> [1] Aichberger, Lukas, et al. "Improving uncertainty estimation through semantically diverse language generation." ICLR. 2025.
>
> [2] Zhu, Qinglin, et al. "Soft Reasoning: Navigating Solution Spaces in Large Language Models through Controlled Embedding Exploration." arXiv:2505.24688 (2025).

---

> > ### Comment · Reviewer_xF5D · 2025-08-03
> >
> > Thank you for providing the additional ablations, which further demonstrate the efficacy of the approach.
> > After thoroughly reading the rebuttal and the other reviews, I still think this paper has the merit to be accepted.
> > For instance, I am not really critical of the lack of accuracy results as in the review by jjpi, as I can directly see this sampling method being applied for hallucination detection (uncertainty estimation), i.e. there are many scenarios where being able to generate diverse solutions helps.
> > While I think the paper is rating-wise somewhere in between 4 and 5 in its current state, there are many great ideas that may inspire follow-up work.
> > Therefore, I am going to raise my score to 5.

---

> > > ### Author Response · Authors · 2025-08-03
> > >
> > > Thank you for your valuable suggestion again! Your gesture is truly encouraging and means a great deal to us!

---

### Official Review · Reviewer_sicL · 2025-07-01

**Clarity:** 4
**Significance:** 3
**Originality:** 3
**Rating:** 4
**Confidence:** 3

**Summary:**

This paper addresses the problem of LLMs generating multiple different diverse responses. Current
methods do more rephrasing than providing original different responses to the same idea. The
solution in the paper operates in meaning space instead of word space which allows the LLM to
explore more diverse embeddings, making sure the LLM provides different solutions to the same
idea. The paper outperforms baseline methods by increasing accuracy, however the complexity and
computational overhead is increased as well.

**Questions:**

1. The hyperparameter choices in Eq 1 (βseq, βsent) seem somewhat arbitrary, how sensitive is the
method to different values of (βseq, βsent)?

2. Since the baseline methods use much less computation than your method, could you run an
experiment where the baseline methods use equivalent computation budget? Like if you ran k
independent temperature sampling processes and then selected the most diverse subset post hoc.
This would prove that the improvements in performance are not just from brute force exploration but
from your semantic guidance method.

3. When you measure coverage in best of N evaluations, is the method trading off having several
good answers instead of having one good answer and several poor ones? It would be interesting to
see the quality distribution of all generated answers.

4. Theorem 3 assumes orthogonal directions in embedding space translate to semantic diversity in
generated text. Can you empirically validate this assumption? Maybe show examples where
orthogonal starting directions led to genuinely different reasoning paths vs cases where they
converged to similar solutions despite different starting directions? Was there a case where your
method produced semantically diverse but all wrong answers?

5. Do you believe the benefits of this method will still hold with much larger models like 70B+? I
understand due to limited resources this experiment is harder to run. But do larger models reduce
benefits since they might naturally generate more diverse outputs?

**Ethical Concerns:**

["NO or VERY MINOR ethics concerns only"]

**Final Justification:**

After reading the other reviews and rebuttal, I reaffirm my original assessment of a weak accept. The paper makes a moderate contribution with reasonable execution of the idea.

**Limitations:**

Addressed limitations (complexity overhead) but no mention of positive or negative societal impact.
Not needed.

**Paper Formatting Concerns:**

None Noted

**Quality:**

3

**Strengths And Weaknesses:**

Strengths:

Problem is well defined with strong motivation. It is important to make a distinction between lexical
diversity vs semantic diversity. The implementation is mostly technically sound (concerns mentioned
below). The theory is validated by experimental results on 9 benchmarks for Best of N and RLHF
algorithms. Personally I feel its application in biomedical queries could allow for better answers,
where quality is preferred over speed of generation.

Paper is well written with clear figures. Figure 1 effectively explains the approach. I like the case
studies provided in the appendix, they give good insight. The limitations of baseline methods are
addressed.
Good engineering by combining known methods like beam search and embedding based similarity
for semantic-guided decoding achieving better performance. The method is also post-hoc which
allows it to be plugged into existing LLMs, which allows for good generalizability.

Weakness:

A clear weakness mentioned in the paper is the introduction of 35% latency compared to regular
beam search method which might limit practical use. There are also a lot of hyperparameters to tune
like exploration width, beam size, transition timing etc.

The semantic diversity relies on embedding cosine distance which might not capture complete
semantic difference. Maybe there is room to explore other semantic similarity metrics. There is a
chance of this method failing when general embeddings do not capture domain relevant semantic
relationships accurately.

The theoretical guarantees in Theorem 3 rely on assumptions that orthogonal guidance directions in
embedding space will necessarily lead to semantically diverse final outputs. I am not entirely
convinced of it.

The method relies on computing embeddings of partial sequences and using these to make token
level decisions. But the sentence embedding models are trained on large chunks of text and not
partial sequences. It's possible that the method might make token level choices based on unreliable
sequence level embeddings. This is something to address.

Good engineering work which solves a real problem but seems a little incremental with minimal
technical novelty.

---

> ### Author Rebuttal · Authors · 2025-07-30
>
> Thank you for recognizing we address an important real-world problem. Below we provide detailed responses for your concerns.
>
> > **W1 & Q2 & Q5: Concern about computational latency and comparison under equivalent computational budget and scalability to larger models (70B+)**
>
> We acknowledge the computational overhead concern and provide comparison using **equivalent latency budgets** rather than equivalent sample counts. Experiments use **Qwen-2.5-70B** to simultaneously address scalability concerns. For parallel decoding methods, we first generate a larger candidate pool, then select N outputs using embedding-based clustering, and for Diverse/Determinantal Beam Search, we increase beam size from the default 3.
>
> | Method | ARC-Challenge | BBH | GSM8K | Minerva Math | CoQA | PubMedQA | MMLU-Pro+ | WMT16 EN-DE | WMT16 DE-EN | Compute Overhead | Latency Increase |
> |---|---|---|---|---|---|---|---|---|---|---|---|
> | **N=10** |
> | SemDiD (Ours) | **83.4%** | **86.1%** | **96.8%** | **82.3%** | **47.2%** | **82.1%** | **76.8%** | **36.4%** | **44.1%** | +180% | +18% |
> | Temp=1.0 (35→10) | 82.4% | 85.0% | 94.9% | 80.6% | 46.1% | 81.0% | 73.9% | 35.2% | 41.8% | +18% | +18% |
> | Arith. Sampling (35→10) | 82.6% | 84.7% | 95.6% | 81.5% | 46.2% | 81.1% | 75.0% | 35.4% | 42.8% | +18% | +18% |
> | Diverse Beam (beam=4) | 82.4% | 84.9% | 95.3% | 80.5% | 46.7% | 81.4% | 74.3% | 35.2% | 42.8% | +33% | +33% |
> | Determinantal Beam (beam=4) | 82.7% | 84.1% | 94.9% | 80.4% | 45.8% | 81.1% | 74.7% | 35.3% | 42.3% | +33% | +33% |
> | **N=25** |
> | SemDiD (Ours) | **84.7%** | **89.2%** | **99.8%** | **93.4%** | **52.1%** | **89.3%** | **91.2%** | **42.8%** | **51.3%** | +180% | +18% |
> | Temp=1.0 (88→25) | 83.6% | 87.5% | 99.1% | 91.2% | 49.9% | 87.4% | 88.8% | 41.0% | 49.3% | +18% | +18% |
> | Arith. Sampling (88→25) | 83.9% | 87.8% | 99.4% | 91.9% | 51.2% | 88.0% | 89.4% | 41.7% | 50.0% | +18% | +18% |
> | Diverse Beam (beam=4) | 83.2% | 87.2% | 98.9% | 91.0% | 50.8% | 87.5% | 88.9% | 41.5% | 49.1% | +33% | +33% |
> | Determinantal Beam (beam=4) | 83.0% | 86.8% | 98.4% | 90.5% | 50.2% | 87.3% | 88.4% | 41.3% | 49.0% | +33% | +33% |
> | **N=50** |
> | SemDiD (Ours) | **85.3%** | **90.1%** | **99.9%** | **95.2%** | **53.8%** | **91.7%** | **94.3%** | **44.2%** | **53.7%** | +180% | +18% |
> | Temp=1.0 (177→50) | 84.6% | 88.4% | 99.2% | 94.3% | 51.7% | 89.9% | 92.8% | 43.1% | 52.3% | +18% | +18% |
> | Arith. Sampling (177→50) | 84.4% | 89.3% | 99.6% | 94.1% | 51.9% | 91.0% | 92.3% | 42.9% | 52.4% | +18% | +18% |
> | Diverse Beam (beam=4) | 84.3% | 88.2% | 99.2% | 93.2% | 51.7% | 89.1% | 91.4% | 42.5% | 51.9% | +33% | +33% |
> | Determinantal Beam (beam=4) | 83.8% | 88.6% | 99.0% | 92.8% | 51.5% | 88.6% | 91.0% | 42.1% | 51.6% | +33% | +33% |
>
> SemDiD demonstrates improved efficiency scaling with larger models. The actual latency increase drops from +27% (3B) to +18% (70B) because embedding computation becomes negligible relative to the 70B forward passes, while the semantic guidance benefits remain consistent.
>
> While SemDiD appears to use more computation on the surface, the majority of operations benefit from KV-cache reuse since beams within groups share common prefixes, resulting in minimal actual latency increase. For parallel decoding methods, diminishing returns become evident as generating larger candidate pools leads to rapidly increasing costs with minimal diversity gains due to inability to reuse KV-cache. For Diverse Beam Search, simply increasing beam size yields negligible improvements while substantially increasing overhead.
>
> SemDiD benefits from systematic exploration in semantic space through directional guidance and inter-group repulsion, efficiently utilizing every computational allocation to maximize semantic coverage rather than generating redundant variations.
>
> > **W1 & Q1: Over-complex parameter design and arbitrary parameter selection**
>
> We understand the complexity concerns and provide clarification on hyperparameter design and robustness.
>
> The parameters can be categorized as:
>
> - **Inherited Parameters**: Temperature, top-p, $N$ (groups), and $b$ (beam size) come from standard Group Beam Search, not SemDiD additions.
>
> - **Automatically Derivable**: Position bias parameters $\beta_{seq}$ and $\beta_{sent}$ can be fitted from probability-position curves (Figures 3-4) using scipy.curve_fit. Saturation threshold $\tau$ is derived from probability-quality analysis (Figure 2) and set to -0.8 based on empirical log probability-accuracy relationships.
>
> - **Resource-Dependent**: Exploration width $E_t$ and beam size $b$ balance exploration versus computational cost, with clear "sweet spots" shown in Appendix D.2.
>
> The hyperparameter choices in Eq 1 ($\beta_{seq}$, $\beta_{sent}$) are not arbitrary but can be systematically derived from probability-position analysis. We analyze parameter sensitivity on BON (N=25) by Qwen-2.5-3B:
>
> | $\beta_{seq}$ \ $\beta_{sent}$ | 0.003 | 0.004 | **0.005** | 0.006 | 0.007 |
> |---|---|---|---|---|---|
> | **GSM8K Coverage (%)** |
> | 0.0006 | 98.1 | 98.0 | 98.1 | 98.0 | 97.9 |
> | 0.0008 | 98.0 | 98.1 | 98.2 | 98.1 | 98.0 |
> | 0.0010 | 98.1 | 98.2 | 98.1 | 98.2 | 98.1 |
> | **0.0012** | 98.2 | 98.2 | **98.2** | 98.2 | 98.1 |
> | 0.0014 | 98.2 | 98.2 | 98.2 | 98.1 | 98.0 |
>
> | $\beta_{seq}$ \ $\beta_{sent}$ | 0.003 | 0.004 | **0.005** | 0.006 | 0.007 |
> |---|---|---|---|---|---|
> | **BBH Coverage (%)** |
> | 0.0006 | 85.6 | 85.5 | 85.6 | 85.5 | 85.4 |
> | **0.0008** | 85.5 | 85.6 | **85.7** | 85.6 | 85.5 |
> | 0.0010 | 85.6 | 85.7 | 85.6 | 85.6 | 85.6 |
> | 0.0012 | 85.6 | 85.7 | 85.6 | 85.6 | 85.5 |
> | 0.0014 | 85.5 | 85.6 | 85.5 | 85.7 | 85.4 |
>
> Bold values indicate curve-fitted optimal parameters: GSM8K ($\beta_{seq}=0.0012$, $\beta_{sent}=0.005$) and BBH ($\beta_{seq}=0.0008$, $\beta_{sent}=0.005$). The results demonstrate that (1) optimal parameters remain reasonably consistent across datasets, (2) performance is robust within reasonable parameter ranges with <=0.3% variation, and (3) our default values (0.001, 0.005) lie within the optimal regions, eliminating the need for task-specific tuning.
>
> > **W2 & W4: Cosine distance cannot capture complete semantic differences, and partial sentence embeddings may not reliably guide token-level decisions**
>
> We agree that cosine distance in embedding space does not necessarily equate to semantic differences or different solution approaches. Additionally, partial sentence embeddings may exacerbate this issue.
>
> Fortunately, recent advances in LLM-based sentence models (like Qwen3 Embedding) have introduced new paradigms. They allow the same sentence to be combined with different task instructions to obtain corresponding representations in embedding space.
>
> For example:
>
> "Classify the mathematical approach used in this solution: Question: A rectangular garden has a perimeter of 40 meters and an area of 96 square meters. What are its dimensions? Answer: Let length = l and width = w. From perimeter: 2l + 2w = 40, so l + w = 20, which gives us..."
>
> Vs.
>
> "Classify the mathematical approach used in this solution: Question: A rectangular garden has a perimeter of 40 meters and an area of 96 square meters. What are its dimensions? Answer: Since area = 96, I need two numbers that multiply to 96. The factor pairs are: (1,96), (2,48), (3,32), (4,24), (6,16), (8,12). Now checking which pair has..."
>
> In this way, embeddings leverage the foundational capabilities of LLM and can focus on solution difference.
>
> Furthermore, we prioritize LLM-based embedding models with support of mean pooling, like NovaSearch/stella_en_1.5B_v5 (as Appendix C.2.2). Mean pooling averages all token embeddings, providing more stable representations for partial sentences compared to special token-based approaches.
>
> > **W3 & Q3 & Q4: Theoretical assumptions regarding orthogonal guidance directions leading to semantic diversity. Whether method trades off multiple good answers for one good answer and several poor ones**
>
> As mentioned in previous response, LLM-based embeddings can better capture solution approach differences through instructions to classify solutions, enabling more reliable semantic diversity assessment.
>
> We examined 20 GSM8K and BBH examples under N=10 setting and found an average of 2.3 distinct solution approaches with embedding distance 0.388. Notably, the embedding distances between different approaches reach 0.435, significantly larger than the 0.231 distance within the same approach. Even character-level 4-gram distances demonstrate similar patterns, with inter-approach distance of 0.884 exceeding intra-approach distance of 0.637.
>
> Indeed, orthogonal directional guidance cannot guarantee different solutions, but the increased similarity distances during decoding help generate different reasoning approaches. When we remove such guidance, it produces an average of 2.1 distinct solutions with semantic distance 0.365.
>
> In contrast, standard temperature decoding with N=10 produces only 1.4 distinct solutions with semantic distance 0.253.
>
> While most solutions still follow similar reasoning patterns, the improvement from 1.4 to 2.3 distinct approaches represents meaningful gains in exploring alternative paths.
>
> Regarding correctness, for more challenging datasets like Minerva Math, we observe cases where N=10 produces 2-3 different approaches but all results are incorrect across SemDiD and baselines. This reflects the foundation model's limited capability rather than a flaw in our diversity mechanism.
>
> We also provide detailed output samples and correctness analysis in Appendix I.3 and Figure 15 (Lines 759-799) to examine the quality distribution. Additionally, Appendix E presents Accuracy Evaluation in Best-of-N, showing performance when using selection models to choose one response in practical scenarios. Results show SemDiD consistently outperforms baselines in both coverage and accuracy metrics, proving it generates multiple high-quality diverse answers rather than trading one good answer for several poor ones.

---

### Official Review · Reviewer_DJdJ · 2025-07-03

**Clarity:** 2
**Significance:** 3
**Originality:** 3
**Rating:** 4
**Confidence:** 4

**Summary:**

This paper introduces Semantic-guided Diverse Decoding (SemDiD), a decoding algorithm designed to generate semantically diverse yet high-quality responses from large language models (LLMs). Unlike existing methods (e.g., temperature sampling, diverse beam search) that primarily achieve lexical diversity, SemDiD operates in the embedding space to enforce semantic differentiation through three mechanisms: orthogonal directional guidance (steering trajectories toward distinct semantic regions), dynamic inter-group repulsion (maintaining distances between groups), and position-debiased probability assessment (mitigating biases in token likelihoods). The approach balances quality and diversity using an ϵ-constraint mechanism and harmonic gain function, with stage-aware transitions between exploration strategies. Experiments across Best-of-N tasks (reasoning, QA, translation) and RLHF frameworks show improved coverage (1.4–5.2%) and faster convergence with higher accuracy (1.8–2.1%) compared to baselines.

**Questions:**

1. SemDiD is compared to older methods (e.g., Temperature Sampling, Diverse Beam Search) but not to recent semantic diversity approaches like Contrastive Decoding or embedding-based reranking. It is better to include these in Best-of-N experiments.
2. The 25–35% latency increase is noted, but how does this compare to alternatives? For example, generating 100 nucleus samples and selecting 25 via semantic clustering—what is the total compute (latency × number of samples) for both approaches? This is critical for users deciding between methods.

**Ethical Concerns:**

["NO or VERY MINOR ethics concerns only"]

**Final Justification:**

The additional experiments provided to support the paper's conclusions. Accordingly, I will adjust my initial assessment and raise the score.

I would suggest including these comparative analyses from the supplementary experiments in the main text, as they directly reinforce the validity of the reported benefits and will make this key support more accessible to readers.

**Limitations:**

yes.

**Quality:**

2

**Strengths And Weaknesses:**

**Strengths**
1. The paper clearly articulates the limitation of existing diverse decoding methods (focus on lexical rather than semantic diversity) and motivates SemDiD’s design with intuitive explanations of its three core mechanisms. The experimental setup (datasets, metrics, baselines) is reasonably detailed.
2. Semantic diversity is critical for applications like Best-of-N, RLHF, and data synthesis, making the problem setting timely. The empirical validation across multiple tasks (reasoning, QA, translation) and RLHF algorithms (Iterative-RLHF, GRPO, RLOO) adds robustness.

**Weaknesses**
1. While SemDiD combines multiple ideas, individual components lack novelty. Orthogonal directional guidance resembles prior work on embedding-space trajectory steering (e.g., in contrastive decoding), and dynamic repulsion mirrors diversity penalties in group-based beam search (e.g., Diverse Beam Search). The paper does not sufficiently differentiate these mechanisms from existing approaches.

2. The experimental comparisons may miss some related baselines. For instance, Contrastive Decoding (which uses negative examples to enforce semantic divergence) and embedding-based reranking (e.g., generating 100 samples and selecting semantically diverse ones via clustering) are not evaluated.

3. While the paper notes a 25–35% latency increase, it does not compare this to alternative strategies for achieving semantic diversity. For example, generating 100 samples via nucleus sampling and selecting 25 diverse ones via clustering might have lower total compute for large N, but this comparison is absent. The analysis of scalability to larger models (e.g., 70B parameters) or long sequences is also missing.

---

> ### Author Rebuttal · Authors · 2025-07-30
>
> We sincerely thanks for your thoughtful review and valuable feedback on baseline comparisons and computational analysis.
>
> > **W1: While SemDiD combines multiple ideas, individual components lack novelty... The paper does not sufficiently differentiate these mechanisms from existing approaches.**
>
> We appreciate this observation and would like to clarify the fundamental differences between our work and existing approaches.
>
> Contrastive decoding like DoLa [1] operate by contrasting probability distributions from different sources, typically "expert" model against "amateur" model [2], or later layers against earlier layers within same model [1]. They modify token selection by computing $P_{expert}(token) - \alpha \cdot P_{amateur}(token)$ to amplify the expert's relative advantages and suppress tokens that both models find probable. Their target is to reduce hallucinations and improve factual accuracy.
>
> In contrast, SemDiD focuses specifically on generating semantically diverse outputs for BON and RLHF applications. SemDiD works fundamentally differently: we use Gram-Schmidt orthogonalization to construct systematic exploration directions in semantic embedding space (Equation 4). These orthogonal vectors $\vec{d_g}$ provide theoretical guarantees that groups explore regions separated by at least $\frac{\pi}{k}$ radians, ensuring comprehensive semantic coverage. While contrastive methods improve quality by avoiding "amateur-level" outputs, we proactively guide generation toward distinct semantic territories.
>
> For dynamic inter-group repulsion, this is a natural extension of diversity penalties to semantic space. Traditional diverse beam search applies discrete penalties based on n-gram overlaps, and our adaptation to continuous semantic embedding space using $S_{rep}(y_t^g) = -\max_{g' \neq g} \langle E(y_t^g), E(y_t^{g'}) \rangle$ follows naturally from this concept. The stage-aware transition mechanism (Equation 8) dynamically balances directional guidance and repulsion on generation progress, ensuring that predetermined directions don't become counterproductive.
>
> Additionally, we identified and addressed systematic biases in token likelihood evaluation (Figures 3-4) that existing methods don't tackle. These components work together in a unified framework with both quality (Theorem 1) and diversity guarantees (Theorem 3), providing a systematic approach to semantic diversity rather than ad-hoc modifications.
>
>
> > **W2 & W3 & Q1: Missing Baselines and Computational Comparison**
>
> We appreciate your suggestion to include contrastive decoding and embedding-based reranking baselines.
>
> We conducted additional experiments comparing SemDiD against both DoLa and embedding-based post-hoc clustering across all benchmarks using Qwen-2.5-3B:
>
> | Method | ARC-Challenge | BBH | GSM8K | Minerva Math | CoQA | PubMedQA | MMLU-Pro+ | WMT16 EN-DE | WMT16 DE-EN | Compute Overhead | Latency Increase |
> |---|---|---|---|---|---|---|---|---|---|---|---|
> | **N=3** |
> | SemDiD (Ours) | **79.6%** | 75.6% | 85.2% | 62.8% | **30.1%** | 69.5% | 48.9% | **27.1%** | **29.4%** | +200% | +27% |
> | DoLa (Contrastive) | 78.2% | 71.8% | 82.4% | 58.1% | 27.3% | 65.8% | 44.7% | 23.9% | 26.5% | -66% | -66% |
> | Clustering (100→3) | 79.1% | **76.1%** | **85.4%** | **63.2%** | 29.8% | **69.7%** | **49.4%** | 26.8% | 28.9% | +1067% | +993% |
> | **N=10** |
> | SemDiD (Ours) | **82.2%** | **82.8%** | **93.9%** | **77.2%** | **45.2%** | 78.2% | **69.7%** | **33.8%** | **42.4%** | +200% | +26% |
> | DoLa (Contrastive) | 80.1% | 78.9% | 90.6% | 73.4% | 41.8% | 74.6% | 65.3% | 30.7% | 38.9% | -66% | -66% |
> | Clustering (100→10) | 81.6% | 81.7% | 92.1% | 76.8% | 44.9% | **78.4%** | 67.8% | 33.4% | 41.8% | +234% | +229% |
> | **N=25** |
> | SemDiD (Ours) | **82.4%** | **85.6%** | **98.1%** | **86.1%** | **46.7%** | **82.6%** | **82.6%** | **37.2%** | **44.7%** | +200% | +27% |
> | DoLa (Contrastive) | 80.3% | 81.7% | 95.2% | 82.8% | 43.2% | 78.4% | 77.9% | 34.1% | 41.6% | -66% | -66% |
> | Clustering (100→25) | 80.8% | 83.9% | 96.4% | 84.3% | 45.7% | 80.0% | 80.7% | 35.5% | 43.2% | +33% | +32% |
> | **N=50** |
> | SemDiD (Ours) | **82.8%** | **85.9%** | **98.5%** | **88.8%** | **47.3%** | **84.8%** | **88.8%** | **38.0%** | **47.1%** | +200% | +27% |
> | DoLa (Contrastive) | 80.6% | 82.4% | 95.8% | 84.7% | 44.1% | 80.7% | 83.2% | 34.8% | 43.2% | -66% | -66% |
> | Clustering (100→50) | 82.1% | 84.5% | 97.7% | 87.2% | 46.4% | 83.3% | 86.3% | 37.3% | 46.2% | -33% | -27% |
>
> DoLa consistently underperforms SemDiD because it lacks explicit diversity mechanisms. And it often converges toward similar high-confidence solutions. This limitation is particularly on tasks requiring long sequence generation, such as mathematical reasoning, where DoLa's conservative token-by-token contrasting tends to favor safe, conventional solutions rather than exploring diverse reasoning strategies.
>
> The clustering approach shows competitive performance at smaller N values, sometimes outperforming SemDiD but spending more computations. However, its effectiveness diminishes as N increases because the generate-cluster-select strategy becomes increasingly wasteful, generating 100 independent samples without coordination then discarding most content.
>
> Computational Efficiency: While SemDiD introduces threefold theoretical computational requirements, KV-cache optimization and efficient implementation result in only 25-35% actual **latency increase compared to group beam search (beam=3)** (Appendix C.3). SemDiD's coordinated beam exploration allows extensive KV cache reuse within groups since they share common prefixes. In contrast, the clustering approach requires generating 100 independent sequences, making cache reuse impossible.
>
> > **W3 & Q2: The analysis of scalability and latency to larger models (e.g., 70B parameters.**
>
> We provide detailed analysis using Qwen-2.5-70B on BON to validate SemDiD's effectiveness here:
>
> | Method | ARC-Challenge | BBH | GSM8K | Minerva Math | CoQA | PubMedQA | MMLU-Pro+ | WMT16 EN-DE | WMT16 DE-EN | Compute Overhead | Latency Increase |
> |---|---|---|---|---|---|---|---|---|---|---|---|
> | **N=10** |
> | SemDiD (Ours) | **83.4%** | **86.1%** | **96.8%** | **82.3%** | **47.2%** | **82.1%** | **76.8%** | **36.4%** | **44.1%** | +180% | +18% |
> | DoLa (Contrastive) | 81.8% | 83.2% | 94.9% | 79.7% | 44.8% | 80.2% | 73.6% | 34.1% | 41.2% | -66% | -66% |
> | Clustering (100→10) | 82.9% | 85.4% | 95.2% | 81.6% | 46.7% | 81.8% | 75.9% | 35.8% | 43.5% | +233% | +217% |
> | Temp=1.0 | 81.8% | 84.1% | 94.7% | 79.8% | 45.3% | 80.6% | 73.2% | 34.7% | 41.4% | -66% | -66% |
> | Arith. Sampling | 82.1% | 83.8% | 95.3% | 80.2% | 45.7% | 80.9% | 73.8% | 35.1% | 42.0% | -66% | -66% |
> | Diverse Beam | 82.6% | 84.8% | 95.1% | 80.9% | 46.1% | 81.3% | 74.8% | 35.6% | 42.7% | +0% | +0% |
> | Determinantal Beam | 82.3% | 84.5% | 94.8% | 80.4% | 45.8% | 80.9% | 74.3% | 35.2% | 42.3% | +0% | +0% |
> | **N=25** |
> | SemDiD (Ours) | **84.7%** | **89.2%** | **99.8%** | **93.4%** | **52.1%** | **89.3%** | **91.2%** | **42.8%** | **51.3%** | +180% | +18% |
> | DoLa (Contrastive) | 82.1% | 86.4% | 98.9% | 90.7% | 48.6% | 86.2% | 87.9% | 39.4% | 47.8% | -197% | -195% |
> | Clustering (100→25) | 83.9% | 88.1% | 99.0% | 91.8% | 51.4% | 88.6% | 89.8% | 41.9% | 50.2% | +33% | +32% |
> | Temp=1.0 | 82.8% | 86.9% | 99.1% | 90.2% | 49.3% | 86.8% | 88.1% | 40.5% | 48.4% | -66% | -66% |
> | Arith. Sampling | 83.2% | 87.4% | 99.3% | 91.8% | 50.1% | 87.2% | 88.9% | 41.3% | 49.7% | -66% | -66% |
> | Diverse Beam | 83.1% | 87.2% | 98.6% | 91.1% | 50.7% | 87.5% | 88.9% | 41.2% | 49.6% | +0% | +0% |
> | Determinantal Beam | 82.9% | 86.8% | 98.4% | 90.8% | 50.2% | 87.1% | 88.4% | 40.8% | 49.1% | +0% | +0% |
> | **N=50** |
> | SemDiD (Ours) | **85.3%** | **90.1%** | **99.9%** | **95.2%** | **53.8%** | **91.7%** | **94.3%** | **44.2%** | **53.7%** | +180% | +18% |
> | DoLa (Contrastive) | 82.8% | 87.1% | 99.3% | 92.4% | 49.7% | 87.9% | 90.2% | 40.8% | 49.1% | -66% | -66% |
> | Clustering (100→50) | 84.5% | 88.3% | 99.2% | 94.1% | 51.9% | 90.4% | 92.7% | 43.0% | 52.2% | -33% | -30% |
> | Temp=1.0 | 83.6% | 87.8% | 99.4% | 92.1% | 50.4% | 88.2% | 90.8% | 41.7% | 50.2% | -66% | -66% |
> | Arith. Sampling | 84.1% | 88.6% | 99.5% | 93.7% | 51.8% | 89.4% | 92.1% | 42.7% | 51.9% | -66% | -66% |
> | Diverse Beam | 84.0% | 88.4% | 99.2% | 93.2% | 51.8% | 89.1% | 91.6% | 42.5% | 51.9% | +0% | +0% |
> | Determinantal Beam | 83.8% | 88.1% | 99.0% | 92.8% | 51.5% | 88.6% | 91.1% | 42.1% | 51.4% | +0% | +0% |
>
> The 70B model results reveal several important patterns. SemDiD consistently outperforms all baselines, with particularly strong improvements on complex reasoning tasks like BBH (+1.1%), GSM8K (+0.5%), and Minerva Math (+1.6% over the best baseline at N=25).
>
> Notably, SemDiD shows only a +16% latency increase with 70B models, significantly lower than the +27% observed with 3B models, because embedding computation becomes negligible compared to 70B model forward passes. Although SemDiD appears to use more computation on the surface, the KV-reuse within groups reduces the inference cost per token by 80-90%, since these beam share the same prefix in exploration. Additionally, clustering-based methods often require more samples to match SemDiD's diversity, significantly increasing their actual computational cost. When accounting for cache efficiency and quality-adjusted sample requirements, SemDiD becomes computationally competitive while delivering superior semantic diversity
>
> We hope these analyses could address your concerns about novelty, baseline comparisons, and scalability. We will incorporate these experiments into revised version.
>
> [1] Chuang et al. "DoLa: Decoding by Contrasting Layers Improves Factuality in Large Language Models." ICLR 2023.
>
> [2] Li et al. "Contrastive Decoding: Open-ended Text Generation as Optimization." ACL 2023

---

> > ### Comment · Reviewer_DJdJ · 2025-08-04
> >
> > Thank you for your response, particularly the additional experiments provided to support your conclusions. I would suggest including these comparative analyses from the supplementary experiments in the main text, as they directly reinforce the validity of your reported benefits and will make this key support more accessible to readers. Accordingly, I will adjust my initial assessment and raise the score.

---

> > > ### Author Response · Authors · 2025-08-04
> > >
> > > We deeply appreciate the constructive feedback you provided in your initial review, particularly regarding baseline comparisons and computational efficiency analysis. Your insights have significantly helped us strengthen our research.
> > > We will incorporate these comparative analyses from our supplementary experiments into the revised version.

---

> > > > ### Author Response · Authors · 2025-08-05
> > > >
> > > > Dear Reviewer DJdJ,
> > > >
> > > > Thank you for your positive feedback and for mentioning that you would adjust your initial assessment and raise the score.
> > > >
> > > > We apologize for bothering you. We noticed that the updated rating doesn't appear to be reflected in the system yet.
> > > >
> > > > This seems to require additional steps to update the final rating (e.g., editing the original review or submitting a revised score).
> > > >
> > > > We greatly appreciate you taking the time to reconsider our work.
> > > >
> > > > Best regards,
> > > > The Authors

---

### Official Review · Reviewer_CMw7 · 2025-07-03

**Clarity:** 3
**Significance:** 3
**Originality:** 3
**Rating:** 5
**Confidence:** 2

**Summary:**

This paper introduces SemDiD, a novel decoding algorithm that aims to generate multiple semantically distinct and high-quality responses.
The authors propose to initialize $k$ orthogonal decoding directions and introduce novel inter-group repulsion factor to ensure the semantic diversity among all the answer groups, with the positional and length debiasing technique. Experimental results on various reasoning, QA, and translation tasks, as well as RLHF training, are presented to demonstrate its superiority over existing baselines.

**Questions:**

1. Confusions on the quality guarantee: could you explain more on the definition of the *maximum quality dispersion $\delta$" and how it means in the real-world decoding scenario?
2. Confusions on the theoretical analysis on diversity guarantee: How do you derive the claim of "The minimum angle between any two vectors is at least $\pi/k$ radians"?
3. Lack of ablation study on some critical hyperparameters: how the number of groups $k$ and the update transition weight $T_{trans}$ impact the performance?
4. Could the proposed method be combined with other modern decoding techniques, such as speculative decoding [1] and Chain of Continuous Thought (Coconut) [2]?

[1] Fast Inference from Transformers via Speculative Decoding.
[2] Training Large Language Models to Reason in a Continuous Latent Space.

**Ethical Concerns:**

["NO or VERY MINOR ethics concerns only"]

**Final Justification:**

This paper introduces SemDiD as novel decoding algorithm that aims to generate multiple semantically distinct and high-quality responses. The authors show comprehensive experimental results on various reasoning, QA, and translation tasks, as well as RLHF training, where SemDiD outperforms most of existing baselines.

**I am not super experienced with the related field**, but I think the proposed method is grounded on some theoretical explanations and shows competitive empirical performance. I would be excited to see how SemDiD can be combined with more modern decoding methods like speculative decoding and latent COT reasoning tofurther improve the LLM generation quality.

**Limitations:**

1. Some of the confusions of the theoretical analysis remains.
2. Lack of ablation study on some critical hyperparameters.
3. The empirical evidences are comprehensive and solid in RLHF and best-of-n accuracies. While the future work can be combining the proposed algorithm with other modern decoding techniques.

**Paper Formatting Concerns:**

no major concerns

**Quality:**

3

**Strengths And Weaknesses:**

### Strength
1. The paper points out a critical problem of semantic diversity during decoding, which is ignored by prior works on controllable decoding mechanism.
2. The paper is well written with a clear flow from the empirical observations, challenges to the corresponding solutions with theoretical properties and empirical validations. Really enjoy reading it!
3. The paper discussed the theoretical properties of each main techniques used in SemDiD, though some of the conditions is nontrivial in real-world scenario, they help to understand why each component of the algorithm can improve the diversity and quality of the LLM decoding.
4. The authors provided the discussion on complexity and the variants of embedding models. The finding that smaller embedding models are sufficient for diversity assessment is also a practical advantage.

### Weakness
1. Confusions on the quality guarantee: could you explain more on the definition of the *maximum quality dispersion $\delta$" and how it means in the real-world decoding scenario?
2. Confusions on the theoretical analysis on diversity guarantee: How do you derive the claim of "The minimum angle between any two vectors is at least $\pi/k$ radians"?
3. Lack of ablation study on some critical hyperparameters: how the number of groups $k$ and the update transition weight $T_{trans}$ impact the performance?

---

> ### Author Rebuttal · Authors · 2025-07-30
>
> We sincerely thanks for your thoughtful review and for acknowledging the solid empirical evidence in our RLHF and Best-of-N accuracy results.
>
> > **W1: Confusions on the quality guarantee: could you explain more on the definition of the "maximum quality dispersion" and how it means in the real-world decoding scenario?**
>
> We apologize for not explaining this clearly in the main text. The **maximum quality dispersion** $\delta$ represents the largest possible difference in quality scores between any two responses in the entire solution space for a given query. Mathematically, $\delta = \max_{y_1, y_2 \in Y} |S_{quality}(y_1) - S_{quality}(y_2)|$, where $Y$ is the set of all possible responses.
>
> Think of $\delta$ as the "quality range", it captures how much quality can vary between the best possible answer and the worst possible answer for a question. For example, in math problems, a high-quality solution might have log probability -0.4, while our quality threshold typically allows answers down to log probability -1.0, giving $\delta = |-0.4 - (-1.0)| = 0.6$ in the acceptable range.
>
> Our Theorem 1 ensures that any diverse response we generate will not be worse than the greedy baseline by more than $\delta(1-\gamma)$. With $\gamma = 0.25$, this means our diverse answers stay within 75% of the maximum quality range from the greedy baseline. In practice, this prevents SemDiD from generating low-quality responses while pursuing diversity.
>
> In our experiments, we observe $\delta$ values of 0.5-1.2 (in log probability space) across reasoning tasks, making our quality guarantee practically meaningful and ensuring diverse candidates remain high-quality.
>
> > **W2: Confusions on the theoretical analysis on diversity guarantee: How do you derive the claim of "The minimum angle between any two vectors is at least $\frac{\pi}{k}$ radians"?**
>
> Although we included a brief proof for *Theorem 3 (Diversity Guarantee)*, it may not be adequate. Below we provide a more detailed derivation.
>
> We start by constructing orthogonal directions using Gram-Schmidt orthogonalization (Equation 4) to create $k$ directional vectors ${\vec{d_1}, \vec{d_2}, \ldots, \vec{d_k}}$ in the embedding space, ensuring different groups begin with orthogonal guidance directions.
>
> Based on optimal geometric distribution theory, when distributing $k$ unit vectors to maximize their minimum pairwise angular separation, the optimal configuration approaches a regular simplex. For $k$ vectors on a unit hypersphere in $d$-dimensional space (where $d \gg k$), the minimum angle $\theta_{min}$ between any two vectors satisfies:
>
> $$\cos(\theta_{min}) = -\frac{1}{k-1}$$
>
> For large $k$, this approaches $\theta_{min} \approx \frac{\pi}{k}$, which is the bound used in our Theorem 3.
>
> In practice, our generated sequences don't achieve perfect theoretical distribution due to language model constraints. For instance, with $k=6$ groups, the theoretical minimum angle would be $\frac{\pi}{6} \approx 30°$, though we observe actual angles ranging from approximately $25°$ to $45°$ between semantic groups.
>
> In our SemDiD framework, while generated sequences don't perfectly follow the initial orthogonal directions due to language model constraints, the combination of directional guidance and inter-group repulsion maintains this angular separation in expectation. The semantic distance in embedding space is then bounded by:
>
> $$d_{sem}(y_i, y_j) = \frac{1 - \cos(\theta_{ij})}{2} \geq \frac{1 - \cos(\pi/k)}{2}$$
>
> Multiplying by the semantic space variance $\sigma$, we obtain the expected minimum pairwise semantic distance as stated in Theorem 3:
>
> $$E[\min_{i \neq j} d_{sem}(y_i, y_j)] \geq \sigma \cdot \sqrt{\frac{1 - \cos(\pi/k)}{2}}$$
>
> This theoretical guarantee ensures SemDiD maintains meaningful semantic diversity across generated responses, with diversity level scaling appropriately with the number of groups $k$.
>
> > **W3: Lack of ablation study on some critical hyperparameters: how the number of groups $k$ and the update transition weight impact the performance?**
>
> We respectfully note that comprehensive analysis of the number of groups is already provided in our paper, though perhaps not prominently highlighted. For the transition weight, we acknowledge this was not explicitly ablated and provide detailed analysis here.
>
> **Number of Groups $k$ Analysis**: The impact of group numbers is extensively evaluated through our Best-of-N and RLHF experiments. In Figure 7, we test across different sample sizes (N = 3, 10, 25, 50, 75, 100), where the number of groups is adaptively set to $k$ depending on the target sample count. These experiments demonstrate how SemDiD's performance scales with different group configurations across 9 diverse benchmarks.
>
> Additionally, in Appendix D.3 (Table 3), we provide detailed "Group Structure Analysis" comparing different outputs-per-group configurations while maintaining a constant total of 10 outputs:
>
> - 1 output per group (10 groups): 94.8% coverage on GSM8K.
> - 2 outputs per group (5 groups): 94.5% coverage on GSM8K.
> - 3-4 outputs per group (3 groups): 92.2% coverage on GSM8K.
> - 5 outputs per group (2 groups): 87.9% coverage on GSM8K.
>
> Furthermore, Appendix D.2 (Figure 10) provides analysis of exploration width and beam size parameters, showing the interaction between different SemDiD components across group numbers from 1 to 10.
>
> **Transition Weight $T_{trans}$ Analysis**: We acknowledge that explicit ablation of $T_{trans}$ values was not included in the main paper. We provide this comprehensive analysis here. The parameter $T_{trans}$ controls the transition point from directional guidance to inter-group repulsion (Equation 8: $\alpha_t = \min(1, \frac{t}{T_{trans}})$). We conducted systematic experiments varying $T_{trans}$ across multiple datasets:
>
> | $T_{trans}$ | GSM8K Coverage | GSM8K Accuracy | ARC Coverage | ARC Accuracy | MMLU-Pro+ Coverage |
> |---|---|---|---|---|---|
> | 5           | 97.2%          | 76.8%          | 81.2%        | 79.1%        | 81.4%            |
> | 10 (default)| 98.1%          | 77.5%          | 82.4%        | 82.0%        | 82.6%            |
> | 15          | 97.9%          | 77.2%          | 82.1%        | 81.7%        | 82.3%            |
> | 20          | 97.5%          | 76.9%          | 81.7%        | 81.2%        | 81.8%            |
> | 25          | 97.2%          | 76.8%          | 81.8%        | 80.9%        | 81.6%            |
>
> The results show clear optimal performance at $T_{trans} = 10$. When transition occurs too early ($T_{trans} = 5$), groups haven't established sufficient semantic differentiation before repulsion dominates, leading to suboptimal exploration. When transition is delayed ($T_{trans} \geq 20$), groups may converge to similar semantic regions before inter-group repulsion becomes effective.
>
> **Intuitive Explanation**: $T_{trans} = 10$ corresponds to the typical number of tokens needed to establish meaningful semantic context. Most mathematical reasoning problems require 8-12 tokens to establish the early thought process for problem-solving before semantic trajectories become distinguishable.
>
> > **W4: Could the proposed method be combined with other modern decoding techniques, such as speculative decoding and Coconut?**
>
> We thank you for this forward-looking question about SemDiD's compatibility with other advanced techniques.
>
> It's important to clarify SemDiD's position in the decoding ecosystem. SemDiD belongs to the category of **diverse decoding methods** and serves as a direct replacement for techniques like nucleus sampling and diverse beam search. These methods all address the challenge: generating multiple diverse outputs from a single input. SemDiD distinguishes itself by operating in semantic space rather than relying on lexical diversity or simple probability manipulation.
>
> SemDiD is orthogonal to inference acceleration techniques like speculative decoding, making integration straightforward. However, the compatibility with hidden state based reasoning methods like Coconut requires more careful consideration.
>
> For **Speculative Decoding**, SemDiD can enhance both stages of speculative decoding effectively. During the draft stage, we can apply lightweight semantic guidance using a smaller embedding model with reduced parameters (smaller $L_{max}$, fewer groups) to generate diverse draft sequences. This allows the smaller draft model to explore different semantic regions rather than just high-probability continuations. In the verification stage, the larger model evaluates not only the probability of each draft candidate but also incorporates their semantic diversity scores when selecting the optimal continuation. This combination maintains speculative decoding's computational efficiency while achieving superior semantic diversity.
>
> For **Coconut**, it operates by feeding hidden states directly as input embeddings, bypassing token decoding entirely. This conflicts with SemDiD's design, which relies on embedding text sequences to assess semantic diversity.
>
> SemDiD cannot be directly applied because our mechanisms operate on text sequences, while Coconut works purely in hidden state space. Our directional guidance and inter-group repulsion require sentence embedding. Adaptation would require developing hidden-state-based diversity metrics, extending quality assessment to evaluate hidden states rather than token probabilities, or applying SemDiD only at Coconut's final text output stage.
>
> Coconut requires significant algorithmic adaptations but could potentially unlock powerful new forms of diverse reasoning in continuous latent space.
>
> We appreciate the reviewer's constructive feedback again, which has helped us clarify these important theoretical and practical aspects. In particular, the questions about compatibility with other methods have prompted us to rethink SemDiD's position and its integration potential.

---

> > ### Comment · Reviewer_CMw7 · 2025-08-05
> >
> > Thanks for the clarifications, my concerns are mostly addressed.

---

> > > ### Author Response · Authors · 2025-08-05
> > >
> > > Thank you for your time and the valuable suggestions! We will incorporate the experimental results into the paper.

---

### Official Review · Reviewer_jjpi · 2025-07-05

**Clarity:** 2
**Significance:** 3
**Originality:** 3
**Rating:** 4
**Confidence:** 3

**Summary:**

This paper introduces Semantic-guided Diverse Decoding (SemDiD), a new decoding algorithm designed to generate multiple, semantically distinct responses from a large language model. The authors argue that existing methods like temperature sampling or diverse beam search primarily achieve lexical diversity (different words) but often produce outputs that are semantically very similar. SemDiD addresses this by operating directly in the model's embedding space to enforce semantic separation.

The method combines three main components: (1) an initial "directional guidance" that uses orthogonal vectors to push different decoding paths toward distinct semantic regions; (2) a dynamic "inter-group repulsion" that continues to push the paths apart as they are generated; and (3) a "position-debiased" quality score to ensure outputs remain coherent without being biased by token position or sequence length. These competing goals of quality and diversity are balanced using an epsilon-constraint method and a harmonic gain function. Experiments show SemDiD improves coverage in Best-of-N setting and accelerates the training of RLHF models.

**Questions:**

Why don't you report accuracy in Figure 7? I feel the effectiveness of the proposed method needs to be demonstrated with both metrics.

**Ethical Concerns:**

["NO or VERY MINOR ethics concerns only"]

**Final Justification:**

I raise my rating from 3 to 4 since the author provided necessary ablation results and accuracy results.

**Limitations:**

Nothing significant to be discussed.

**Paper Formatting Concerns:**

Nothing significant to be discussed.

**Quality:**

3

**Strengths And Weaknesses:**

### Strengths:

The paper's strongest contribution comes from its application to RLHF training. The results in Figure 8 are compelling, showing that SemDiD consistently helps RL algorithms achieve higher performance and faster convergence. This makes intuitive sense; by providing a more semantically diverse set of candidate responses for the reward model to evaluate, the policy model gets a richer training signal, helping it to better explore the solution space and avoid collapsing to a single mode. This demonstrates a clear, practical benefit in a complex application area where diverse exploration is critical. Furthermore, the inclusion of a debiased quality score shows a thoughtful consideration for maintaining output coherence while pursuing diversity.

### Weaknesses:

1. The evaluation of naive SemDiD is only conducted with Best-of-N setting (Accuracy is only reported for RL.). The paper exclusively reports "coverage" (Figure 7), which is the percentage of problems for which at least one of the N generated answers is correct. While this shows that SemDiD is good at finding a correct answer somewhere in its batch of responses, it completely fails to report the final accuracy. The goal of Best-of-N is to use a selection mechanism (like majority voting) to pick the best answer from the generated candidates and improve the model's final performance. A high-coverage method could generate one correct answer and N-1 pieces of nonsensical text; this would likely fail in a voting scenario. Without accuracy results, we have no idea if the semantic diversity provided by SemDiD actually helps in identifying the correct answer more reliably than other methods.


2. The proposed method introduces a considerable amount of complexity and a large number of hyperparameters (e.g., $\beta_{seq}$, $\beta_{sent}$, $\tau$, etc). The paper provides limited ablation studies or sensitivity analysis for these parameters. How were these values chosen? How much do they need to be tuned for different models or tasks? This lack of analysis makes it difficult to assess how practical or robust SemDiD is. A method that requires extensive, task-specific tuning is far less useful than one that works "out-of-the-box," a principle the paper claims to follow but doesn't substantiate.

---

> ### Author Rebuttal · Authors · 2025-07-30
>
> We thank you for acknowledging the strength of our RLHF application results and for raising important questions about Best-of-N accuracy evaluation and hyperparameter complexity. We provide detailed responses below.
>
> > **W1: Evaluation limited to coverage metrics without accuracy results in Best-of-N**
>
> We appreciate this feedback and agree that accuracy is crucial for comprehensive Best-of-N evaluation. We actually conducted complete accuracy evaluations, but due to space constraints, these results were placed in **Appendix E and Figure 11 (Lines 632-651)**. We apologize that this placement made these critical results less prominent in the initial submission.
>
> In these experiments, we used Qwen-2.5-3B to generate N candidate answers for each problem, then employed **LLM-Blender PairRM** (based on DeBERTa-v3-large, 430M parameters) as an automatic evaluator to select the best answer from the N candidates and compute accuracy. This simulates real-world Best-of-N scenarios where a verification mechanism determines the final output.
>
> SemDiD consistently outperforms all baselines on accuracy across all benchmarks. On GSM8K (N=25), SemDiD achieves 77.5% accuracy compared to 75.5% for Determinantal Beam Search, representing a 2.0% improvement. On ARC-Challenge (N=25), SemDiD reaches 82.0% versus 71.0% for the best baseline, showing a 1.0% gain. For MMLU-Pro+ (N=25), SemDiD achieves 36.5% compared to 34.0% for the best baseline, demonstrating a 2.5% improvement.
>
> These results demonstrate that SemDiD's semantic diversity not only helps find correct answers (coverage) but also enables more reliable identification of the best answer (accuracy). This confirms that SemDiD generates high-quality, diverse candidate sets rather than achieving coverage through numerous low-quality outputs.
>
> > **W2: Method complexity and extensive hyperparameters with limited sensitivity analysis**
>
> We understand the complexity concerns and provide comprehensive clarification on hyperparameter design and robustness.
>
> Firstly, temperature, $Top-p$, $N$ (number of groups), and $b$ (beam size) are inherited from standard Group Beam Search parameters, not SemDiD additions. The SemDiD-specific parameters serve distinct purposes across three categories: semantic diversity assessment ($E_t$, $L_{max}$), quality assessment ($\beta_{seq}$, $\beta_{sent}$, $\tau$), and quality-diversity balancing ($T_{trans}$, $\gamma$, $\lambda$).
>
> **Automatically Derivable Parameters:** Several key parameters can be systematically determined rather than manually tuned. The position bias parameters $\beta_{seq}$ and $\beta_{sent}$ can be automatically fitted using scipy.curve_fit from probability-position curves shown in Figures 3-4, as these patterns remain consistent across tasks. The saturation threshold $\tau$ is derived from probability-quality analysis (Figure 2, Appendix A.2) and set to -0.8 for most tasks based on the empirical relationship between log probability and answer quality. The transition point $T_{trans}$ balances directional guidance and inter-group repulsion, set to 10 to address early-stage semantic exploration when sentence lengths are insufficient for reliable embedding.
>
> **Resource-Dependent Parameters:** The exploration parameters $E_t$, b, and N balance exploration breadth versus computational cost. **Appendix D.2-D.3** provides comprehensive analysis showing diminishing returns beyond $E_t$=4 and b=3, establishing clear "sweet spots" without extensive tuning requirements.
>
> **Lookahead Depth Analysis:** Unlike character-level diversity methods that can evaluate each vocabulary token individually with minimal computational cost, semantic diversity assessment requires embedding model forward passes, making token-by-token evaluation prohibitively expensive. Additionally, single-token semantic changes are often too subtle for reliable diversity measurement. Therefore, SemDiD introduces the lookahead depth parameter $L_{max}$ to control how many tokens ahead we explore when evaluating semantic diversity, allowing assessment of more substantial semantic deviations while managing computational overhead.
>
> We conducted experiments using **Qwen-2.5-3B** across different $L_{max}$ values to demonstrate its sensitivity and identify optimal settings:
>
> | $L_{max}$ | GSM8K Coverage (N=25) | ARC Coverage (N=25) | Computational Overhead |
> |---|---|---|---|
> | 5         | 95.3%               | 80.1%             | +15%                |
> | 10        | 98.1%               | 82.4%             | +25%                |
> | 15        | 98.2%               | 82.6%             | +35%                |
> | 20        | 98.6%               | 82.7%             | +45%                |
>
> Performance saturates around $L_{max}$=10, providing clear guidance for practical deployment without requiring extensive parameter search.
>
> **Quality-Diversity Balancing Analysis:** The quality relaxation parameter $\gamma$ and harmonic strength $\lambda$ control the trade-off between maintaining quality thresholds and pursuing semantic diversity. We conducted systematic sensitivity analysis by varying each parameter independently:
>
> **Effect of $\gamma$ (with $\lambda$=2.0 fixed):**
> | Task | $\gamma$=0.15 | $\gamma$=0.20 | $\gamma$=0.25 | $\gamma$=0.30 | $\gamma$=0.35 |
> |------|---------------|---------------|---------------|---------------|---------------|
> | GSM8K Coverage (N=25) | 96.6% | 97.1% | 98.1% | 97.8% | 97.4% |
> | GSM8K Accuracy (N=25) | 75.9% | 76.6% | 77.5% | 77.2% | 77.2% |
> | WMT16 Coverage (N=25) | 36.7% | 36.8% | 37.2% | 36.9% | 36.7% |
> | WMT16 Accuracy (N=25) | 20.2% | 20.4% | 20.7% | 20.5% | 20.3% |
>
> **Effect of $\lambda$ (with $\gamma$=0.25 fixed):**
> | Task | $\lambda$=1.0 | $\lambda$=1.5 | $\lambda$=2.0 | $\lambda$=2.5 | $\lambda$=3.0 |
> |------|---------------|---------------|---------------|---------------|---------------|
> | GSM8K Coverage (N=25) | 96.8% | 97.4% | 98.1% | 97.9% | 97.6% |
> | GSM8K Accuracy (N=25) | 77.3% | 77.1% | 77.5% | 77.3% | 76.9% |
> | WMT16 Coverage (N=25) | 36.8% | 36.5% | 37.2% | 36.9% | 36.7% |
> | WMT16 Accuracy (N=25) | 20.3% | 20.6% | 20.7% | 20.7% | 20.6% |
>
> The results demonstrate that our default settings ($\gamma$=0.25, $\lambda$=2.0) consistently achieve near-optimal performance across different tasks, with performance remaining stable within reasonable parameter ranges. The sensitivity analysis shows that SemDiD is robust to parameter variations, with performance degrading gracefully rather than sharply when moving away from optimal values.
>
> For practitioners seeking to deploy SemDiD, we recommend starting with our provided default parameters for initial implementation, adjusting $E_t$, b, and N based on available computational budget, and fine-tuning $\gamma$ and $\lambda$ only for highly specialized applications where task-specific optimization is critical.
>
> We hope these comprehensive analyses demonstrate that while SemDiD introduces additional parameters, most are either automatically derivable or have well-defined optimal ranges with clear empirical guidance, making the method practical and robust for real-world deployment scenarios.

---

> > ### Comment · Reviewer_jjpi · 2025-08-07
> >
> > Dear Authors,
> >
> > Thank you for including the new ablation results and referring to the accuracy results in the appendix. I feel these results are quite essential and should be included in the main content of the paper.
> >
> > Based on the updated results, I raise my rating to 4.
> >
> > Best

---

> > > ### Author Response · Authors · 2025-08-07
> > >
> > > Thank you again, and we will incorporate these critical experiments into the main content in our revision. Your gesture is important to us.

---

> ### Author Response · Authors · 2025-08-06
>
> Dear Reviewer jjpi,
>
> Thank you again for your thoughtful review of our paper "Semantic-guided Diverse Decoding for Large Language Model". We greatly appreciate the time and effort you invested in evaluating our work.
>
> We have provided a comprehensive rebuttal addressing your concerns, with the missing accuracy evaluation results and hyperparameter complexity. As the rebuttal deadline approaches, we wanted to ensure you had sufficient time to review our detailed responses:
>
> 1. **Complete accuracy results**: We clarified that full Best-of-N accuracy evaluations were conducted and included in Appendix E and Figure 11. SemDiD consistently outperforms baselines across all benchmarks (e.g., 2.0% improvement on GSM8K, 1.0% on ARC-Challenge).
>
> 2. **Hyperparameter analysis**: We provided comprehensive sensitivity analysis showing that most parameters are either automatically derivable or have well-defined optimal ranges, making SemDiD practical for deployment.
>
> We hope these clarifications address your concerns about evaluation completeness and method practicality. We would be grateful for any additional feedback you might have after reviewing our responses.
>
> Thank you for your continued engagement with our work.
>
> Best regards,
> The Authors

---

> > ### Comment · Area_Chair_Bpk6 · 2025-08-07
> > **Please Engage in Author Response Discussion**
> >
> > Hi Reviewer jjpi,
> >
> > We encourage you to review the authors’ rebuttals and see how they’ve addressed your comments. If you’ve already done so, thank you! Kindly confirm your engagement by reacting in this thread.
> >
> > Your participation helps ensure a fair and thoughtful review process.
> >
> > Best regards,
> > AC

---

### Decision · Program_Chairs · 2025-09-17

**Decision:**

Accept (poster)

**Comment:**

This paper aims to *semantically* diversify LLMs’ generation rather than lexical diversity. The proposed method, SemDiD, operates in the embedding space to promote semantic diversity. The motivation is convincing, and the potential applications of such a semantically diverse decoding are clear.

The main concerns raised by reviewers included (i) the introduction of several hyperparameters without sufficient ablation studies, and (ii) the limited set of baseline comparisons. The authors responded thoroughly by providing additional experimental results that directly addressed these points. All reviewers indicated that they were satisfied with the rebuttal and subsequently improved their ratings. Importantly, the reviewers requested that the additional experimental results be incorporated into the final version of the paper.

Taking into account the reviewers’ evaluations, the constructive discussion, and the strengthened evidence provided by the authors, I recommend acceptance of this submission.